# Intervene-All-Paths: Unified Mitigation of LVLM Hallucinations across Alignment Formats

**Jiaye Qian**[1,2*] **Ge Zheng**[2*] **Yuchen Zhu**[2] **Sibei Yang**[1†]

[1]School of Computer Science and Engineering, Sun Yat-sen University
[2]ShanghaiTech University
Project Page: https://github.com/SooLab/AllPath

## Abstract

Despite their impressive performance across a wide range of tasks, Large Vision-Language Models (LVLMs) remain prone to hallucination. In this study, we propose a comprehensive intervention framework aligned with the transformer's causal architecture in LVLMs, integrating the effects of different intervention paths on hallucination. We find that hallucinations in LVLMs do not arise from a single causal path, but rather from the interplay among image-to-input-text, image-to-output-text, and text-to-text pathways. For the first time, we also find that LVLMs rely on different pathways depending on the question–answer alignment format. Building on these insights, we propose simple yet effective methods to identify and intervene on critical hallucination heads within each pathway, tailored to discriminative and generative formats. Experiments across multiple benchmarks demonstrate that our approach consistently reduces hallucinations across diverse alignment types.

## 1 Introduction

Large Vision-Language Models (LVLMs) [36, 37, 3, 4, 86, 6, 8, 9, 73, 78] have emerged as foundational models [53, 11, 51, 87, 13] for general-purpose visual understanding [67, 66, 68, 69, 70, 84, 71, 32, 52, 21] and reasoning [48, 40, 56, 64, 57, 81], demonstrating strong capabilities across a broad range of open-world tasks [19, 48, 58, 49, 50, 20, 31, 34, 12]. They are able to handle diverse question formats [5], from binary judgments and multiple-choice queries to open-ended captions and dialogues. However, despite these advances, LVLMs often suffer from hallucination, generating textual outputs that are inconsistent with the visual input, such as images [30, 80, 23, 18, 41]. This issue underscores a fundamental limitation of LVLMs and raises concerns about user trust and the potential for misinformation. Therefore, tackling hallucination has become a growing research focus, with increasing efforts to uncover its causes and mitigate its impact to enhance the reliability of LVLMs.

The causes of hallucination in LVLMs are inherently more complex due to their foundation on text-only LLMs. Broadly, hallucinations may be inherited from the underlying LLMs or stem from vision-specific components and training. More specifically, several studies have examined the concrete factors and proposed corresponding mitigation techniques. Contrastive decoding attributes hallucination to over-reliance on statistics and unimodal language priors, mitigating it by calibrating LVLM output distributions via comparisons with logits from distorted visual inputs [28, 27, 61]. Intervention-based techniques diagnose internal attention weights or heads in LVLMs and directly intervene, enabling correction during inference with a single forward pass and no additional computational cost [39, 72, 83]. Among them, weight intervention [39, 83] enhances visual grounding by

---

*Equal contribution. Work done during Jiaye's internship at SYSU.
†Corresponding author is Sibei Yang.

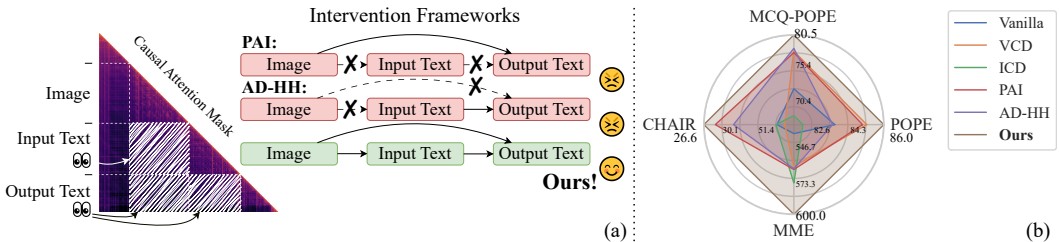

Figure 1: Left: Our AllPath intervention frameowrk comprehensively mitigates hallucinations from image-to-input-text, image-to-output-text, and text-to-text paths. Right: AllPath achieves significant performance improvements over the baselines across all benchmarks.

reinforcing attention to the image, while head intervention [72] mitigates text-dominant behavior by suppressing "lazy" heads that favor text and resemble those in the base LLM after tuning.

In this paper, we explore the causes of hallucination in LVLMs by analyzing their internal components, following the intervention-based line of work. As [72] shows that attention heads have a greater influence on hallucination than MLPs, we focus on heads as the target components. We begin with an empirical observation that motivates our comprehensive intervention framework, which is validated through the identification of critical heads, counterfactual edits, correlation analysis, and empirical evaluation. This process comprises four stages, detailed in below. Notably, our primary analysis focuses on LLaVA-v1.5-7B [36], with additional results for Qwen-VL-Chat [3] and Qwen2.5-VL-7B/72B [4] are provided in the Appendix C.

First, ***we propose a comprehensive intervention framework aligned with the transformer's causal architecture in LVLMs***, based on a thorough evaluation of existing hallucination studies. As shown in Fig. 1(b), a single hallucination mitigation method often struggles to achieve consistently strong performance across different benchmarks [29, 45, 17]. Focusing on intervention methods [27, 61, 39, 72], we observe that they typically intervene along a single causal path—either from input image to output text (PAI in Fig. 1(a)) or from input text to output text (AD-HH in Fig. 1(a))—without considering the system holistically. As a result, certain methods [39, 72] perform better on benchmarks [45] that primarily rely on visual grounding, while others [27] are more effective on benchmarks [29] emphasizing textual coherence. Inspired by this, we propose a comprehensive intervention framework aligned with the input-output sequence and causal transformer architecture of LVLMs, as shown in Fig. 2(Ours). This framework enables joint intervention of the image-to-output-text and image-to-input-text-to-output-text causal pathways. Notably, although the input text is provided rather than generated, its hidden representations are influenced by the input image.

Second, ***we identify the heads in LVLMs that govern distinct intervention pathways within the comprehensive framework by introducing two novel methods***, each targeting the image-to-text and text-to-text causal paths, respectively (detailed in Sec. 2). Although the framework involves multiple causal paths, they fall into two main types: text-to-text (input to output text) and image-to-text (from image to input or output text). For the former type, as textual output logits are primarily influenced by preceding text tokens [38], we identify the path's heads by measuring their differential impact on the logits of hallucinated versus non-hallucinated tokens. For the latter type, the image's influence on text is reflected in the attention contributions from image tokens to text tokens. Path heads are identified by assessing whether they attend to image tokens that are semantically aligned with the corresponding text. Notably, both of our methods are significantly faster than previous approaches for identifying hallucination-related heads, detailed in Sec. 2.4

Third, we validate the impact and roles of the identified heads both within individual causal paths and in their combined pathways, revealing key insights into the comprehensive intervention framework (detailed in Sec. 3). We find that:

- ***Text-to-text intervened heads primarily contribute to LVLM alignment, particularly in adhering to instruction-following formats.*** Heads are highly correlated and often overlap in questions with similar formats, even across different benchmarks, but differ significantly in more distinct formats (e.g., binary vs. open-ended captioning).

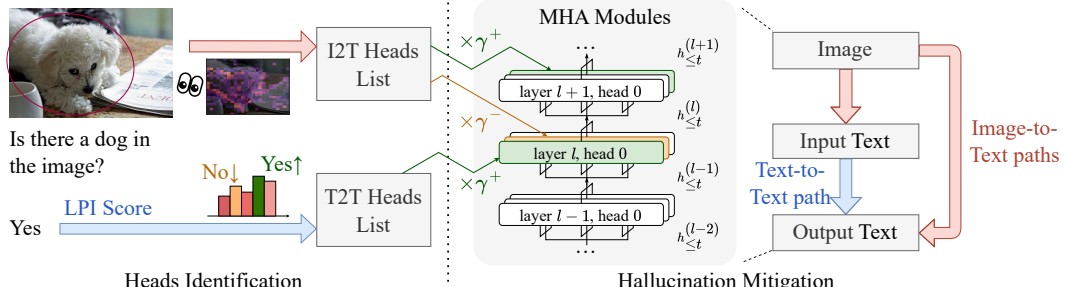

Figure 2: The overview of our proposed AllPath. AllPath first identifies the most critical text-to-text and image-to-text heads contributing to hallucinations using the Log Probability Increase (LPI) score and the ratio of key object attention to total image attention. Then, by applying adaptive heads interventions, AllPath mitigates hallucinations by manipulating the casual pathways in LVLMs.

- ***The intervened heads along the image-to-input-text and image-to-output-text paths differ***, despite both exhibiting stronger attention to image tokens than to text tokens. This suggests that, although both paths "see" the image, the heads governing input and output text function independently.
- ***LVLMs are "smarter" than expected, adaptively selecting the most suitable combined pathways based on the question type***—such as image-to-output-text, image-to-input-text-to-output-text, or both—depending on the relative visual and textual demands of the question.

Finally, ***we propose a simple yet effective intervention method for mitigating hallucination***, grounded in our earlier findings. It adaptively selects the appropriate pathway(s) within the comprehensive intervention framework based on question type and intervenes on the corresponding heads. Experiments demonstrate that our method consistently improves performance across diverse benchmarks while maintaining high efficiency.

In summary, our contributions are multifold: (1) To the best of our knowledge, we are the first to propose a multi-path hallucination intervention framework for LVLMs, going beyond prior single-path approaches. (2) We introduce two novel methods to identify critical heads across different causal pathways. (3) Through counterfactual edits and correlation analysis on attention heads, we uncover several novel findings—reported for the first time—that offer insights into the internal mechanisms of LVLMs. (4) Built upon this framework, we demonstrate that simple interventions on selected heads yield consistent improvements across benchmarks with diverse question and alignment formats, while maintaining high efficiency.

## 2 Probing image-to-text and text-to-text heads along causal paths

Building on the intervention framework, we propose two lightweight and plug-and-play methods to identify critical attention heads associated with each type of causal pathway—text-to-text and image-to-text. An overview of the our probing method is illustrated in Fig. 2.

### 2.1 Preliminaries

We consider an LVLM composed of an image encoder, a modality connector and a base LLM with $L$ transformer layers. At each generation time step $t$, the LVLM first encodes the input image $I$, text prompt $P$, and the previously generated response tokens $R_{\leq t}$ into initial embeddings $h_{\leq t}^{(0)}$. This embedding sequence is then progressively updated through the transformer layers, with the hidden states at layer $l$ updated as follows:

$$h_{\leq t}^{(l)} = h_{\leq t}^{(l-1)} + H_{\leq t}^{(l)} + F_{\leq t}^{(l)}$$

where $H_{\leq t}^{(l)}$ and $F_{\leq t}^{(l)}$ are the outputs of the multi-head attention (MHA) module and the feed-forward network (FFN), respectively. The final hidden state at the current time step, $h_t^{(L)}$, is then used to predict the next token $r_{t+1}$, which is sampled from:

$$\mathbb{P}(r_{t+1} \mid h_t^{(L)}) = \text{softmax}(\text{Unembed}(h_t^{(L)}))$$

| **Algorithm 1:** Get Scores Set for Short Answering Questions | **Algorithm 2:** Get Scores Set for Open-ended Questions |
|---|---|
| **for** *each sample, each time step $t$ in the model generation process* **do**
  **if** *$t$ is the first time step in generation* **then**
    $\mathcal{B}_t^+ \leftarrow$ correct answer token of the question;
    $\mathcal{B}_t^- \leftarrow$ incorrect answer token(s) of the question;
  **else**
    $\mathcal{B}_t^+, \mathcal{B}_t^- \leftarrow \emptyset$; | **for** *each sample, each time step $t$ in the model generation process* **do**
  $\mathcal{B}_t^+, \mathcal{B}_t^- \leftarrow \emptyset$;
  **if** *$r_t$ is judged as hallucination* **then**
    $\mathcal{B}_t^- \leftarrow \{r_t\}$;
  **else if** *$r_t$ is judged as non-hallucination* **then**
    $\mathcal{B}_t^+ \leftarrow \{r_t\}$; |

In this paper, we focus on the MHA module in transformer layers. Let $N$ denote the number of attention heads employed in each layer. For a specific head indexed by $(l, n)$, where $l \in \{1, 2, \ldots, L\}$ is the layer index and $n \in \{1, 2, \ldots, N\}$ is the head index within that layer, we denote the corresponding query, key, value, and output projection matrices as $W_Q^{(l,n)}$, $W_K^{(l,n)}$, $W_V^{(l,n)}$, and $O^{(l,n)}$, respectively. Given the layer input $h_{\leq t}^{(l-1)}$, the output $H_{\leq t}^{(l,n)}$ is computed as:

$$A_{\leq t}^{(l,n)} = \mathrm{softmax}\left(\frac{W_Q^{(l,n)} h_{\leq t}^{(l-1)} (W_K^{(l,n)} h_{\leq t}^{(l-1)})^\top}{\sqrt{d_k}}\right)$$

$$H_{\leq t}^{(l,n)} = A_{\leq t}^{(l,n)} \cdot W_V^{(l,n)} h_{\leq t}^{(l-1)} \cdot O^{(l,n)}$$

where $\sqrt{d_k}$ is the scaling factor and $A_{\leq t}^{(l,n)}$ is the attention weight for head $(l, n)$. The final output of MHA at layer $l$ is the sum of $N$ head output, and can be expressed as $H_{\leq t}^{(l)} = \sum_{i=1}^{N} H_{\leq t}^{(l,i)}$.

## 2.2 Discovery of crucial attention heads for the text-to-text path

Given that the textual output relies heavily on preceding text tokens, output probability distribution reflects the model's confidence in hallucinated content, we refine the Log Probability Increase (LPI) score [76, 77] to quantitatively assess the influence of individual model components in promoting or mitigating hallucinations. Specifically, for a generated token $r_t$ at time step $t$, the LPI score of attention head $(l, n)$ is calculated as:

$$\mathrm{logProb}_\uparrow^{(l,n)}(r_t) = \log \mathbb{P}(r_t \mid h_t^{(l-1)} + H_t^{(l,n)}) - \log \mathbb{P}(r_t \mid h_t^{(l-1)})$$

which measures the contribution of head $(l, n)$ to token $r_t$. To further capture the effect of the head over a broader set of tokens, we extend the definition of LPI score by considering a candidate vocabulary set $\mathcal{B}_t$ at time $t$, and compute the modified score as:

$$\mathrm{logProb}_\uparrow^{(l,n)}(\mathcal{B}_t) = \log \sum_{b \in \mathcal{B}_t} \mathbb{P}(b \mid h_t^{(l-1)} + H_t^{(l,n)}) - \log \sum_{b \in \mathcal{B}_t} \mathbb{P}(b \mid h_t^{(l-1)})$$

At the decoding step $t$, we designate a hallucination set $\mathcal{B}_t^-$ and a non-hallucination set $\mathcal{B}_t^+$. Specifically, for short answering questions, we prompt the model to produce a direct response and analyze the first decoding step, assigning the correct answer to $\mathcal{B}_t^+$ and all incorrect alternatives to $\mathcal{B}_t^-$. For open-ended questions, we focus on positions that have been identified as hallucinated or non-hallucinated. The corresponding token is assigned exclusively to either $\mathcal{B}_t^-$ or $\mathcal{B}_t^+$, with the other set left empty (see Alg. 1 and 2 for details). Then, for each attention head $(l, n)$, we compute the LPI score on both sets, and then average the scores over all decoding time steps across all samples:

$$S_{\mathrm{T2T}}^{(l,n),-} = \mathbb{E}_{\mathcal{B}_t^- \neq \emptyset}[\mathrm{logProb}_\uparrow^{(l,n)}(\mathcal{B}_t^-)], \qquad S_{\mathrm{T2T}}^{(l,n),+} = \mathbb{E}_{\mathcal{B}_t^+ \neq \emptyset}[\mathrm{logProb}_\uparrow^{(l,n)}(\mathcal{B}_t^+)]$$

We then define the **Text-to-Text (T2T) Score** for each head as the difference between the two:

$$S_{\mathrm{T2T}}^{(l,n)} = S_{\mathrm{T2T}}^{(l,n),+} - S_{\mathrm{T2T}}^{(l,n),-}$$

where a smaller value indicates a stronger tendency of head $(l, n)$ to promote hallucinations.

## 2.3 Discovery of crucial attention heads for the image-to-text path

In addition to the textual context, visual information is another crucial factor influencing the model's responses. To identify Image-to-Text (I2T) attention heads, we first select a set of tokens which are most important for grounding the image, denoted as $\mathcal{T}_{\text{I2T}}$. Specifically, since LVLMs are causal models that later tokens integrate information from earlier ones through attention mechanism, we focus only on the *first occurrence* of each object during the question-answering process, either the input text $P$ or the output response $R_{\leq t}$, as these initial mentions are expected to be grounded in the relevant image regions rather than inferred from prior textual context. This makes these selected tokens especially useful for probing I2T heads.

To distinguish whether the corresponding objects of these tokens are present in the image, we then partition $\mathcal{T}_{\text{I2T}}$ into two disjoint subsets: $\mathcal{T}_{\text{I2T}}^{+}$ and $\mathcal{T}_{\text{I2T}}^{-}$, where $\mathcal{T}_{\text{I2T}}^{+}$ contains tokens whose objects are present in the image, and $\mathcal{T}_{\text{I2T}}^{-}$ contains tokens whose objects are absent. When $t \in \mathcal{T}_{\text{I2T}}^{+}$, we expect the attention to be both strong and concentrated within the relevant region. Accordingly, we define the positive score $S_{\text{I2T}}^{(l,n),+}$ as the average summed attention scores within the aligned region $M_r$ across all image tokens. Conversely, for $t \in \mathcal{T}_{\text{I2T}}^{-}$, the attention is expected to be weaker and more diffuse. In this case, we define the negative score $S_{\text{I2T}}^{(l,n),-}$ as the average summed attention scores over the entire image region. Formally, we compute:

$$S_{\text{I2T}}^{(l,n),+} = \mathbb{E}_{r \in \mathcal{T}_{\text{I2T}}^{+}} \Big[ \sum_{i \in M_r} A_{r,i}^{(l,n)} \Big], \qquad S_{\text{I2T}}^{(l,n),-} = \mathbb{E}_{r \in \mathcal{T}_{\text{I2T}}^{-}} \Big[ \sum_{i \in I} A_{r,i}^{(l,n)} \Big]$$

Finally, similar to Text-to-Text Score, we obtain the **Image-to-Text (I2T) Score** for each head as:

$$S_{\text{I2T}}^{(l,n)} = S_{\text{I2T}}^{(l,n),+} - S_{\text{I2T}}^{(l,n),-}$$

## 2.4 Efficiency analysis of head probing methods

Previous work identifies hallucination-related attention heads via either a zero-out strategy [72] or training-based methods [15]. However, both approaches incur substantial computational overhead. The zero-out strategy requires ablating each attention head at every transformer layer to estimate its contribution to hallucination, necessitating a full forward pass at each generation time step $t$ for every head. Meanwhile, training-based methods involve training an additional classifier on a large annotated dataset.

In contrast, our method computes Text-to-Text (T2T) Score and Image-to-Text (I2T) Score for each attention head based solely on local behavior—the log-probability increase at that layer or attention shifts within the layer—without requiring full inference. This allows us to obtain all heads' T2T and I2T scores with only a single forward pass, significantly enhancing both efficiency and practicality.

# 3 Analysis and discussion on identified heads in different paths

In this section, we determine the specific roles of the heads we identified by analyzing their statistical information and behaviors. First, in Sec. 3.1, we investigate the behaviors of the heads identified by our two methods and examine how they contribute to the text-to-text and image-to-text causal pathways, respectively. Subsequently, in Sec. 3.2, we show that LVLMs adaptively select different pathways across benchmarks, and that pathway choice significantly affects hallucination occurrence and mitigation across question types.

## 3.1 Functional interpretation of the identified heads

**Text-to-text causal path is related to the alignment.** To investigate the relationships between the heads identified in different datasets, we independently identify heads within each dataset and analyze the correlations between them. Specifically, we focus on comparing the heads identified in the POPE [29] COCO [33] adversarial dataset with those found in other datasets. For each dataset, we rank the Importance Scores $S_{\text{T2T}}^{(l,n)}$ and $S_{\text{I2T}}^{(l,n)}$ for each head. We then pairwisely plot the ranking positions of each head on a two-dimensional graph and calculate the linear regression correlation

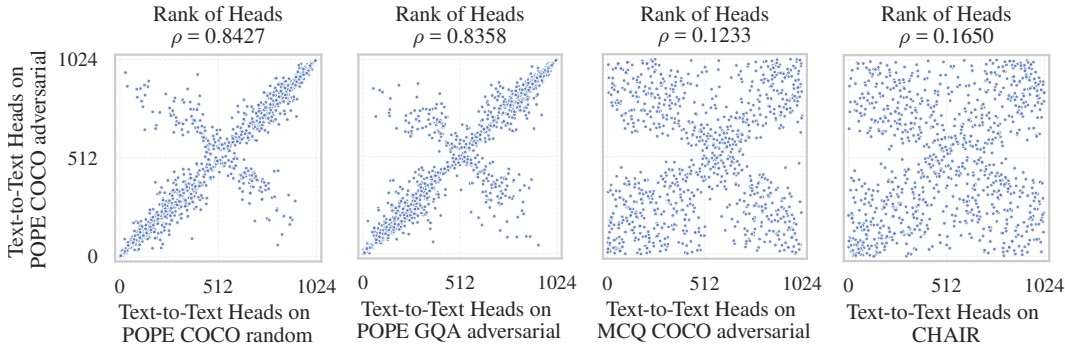

Figure 3: Visualization of the rank distributions of text-to-text heads extracted from different datasets. $\rho$ denotes the correlation coefficient; a higher value (i.e., points concentrated along the diagonal) indicates greater similarity between the two sets of heads. Heads identified from datasets with similar question–answer alignment formats exhibit strong correlation and substantial overlap.

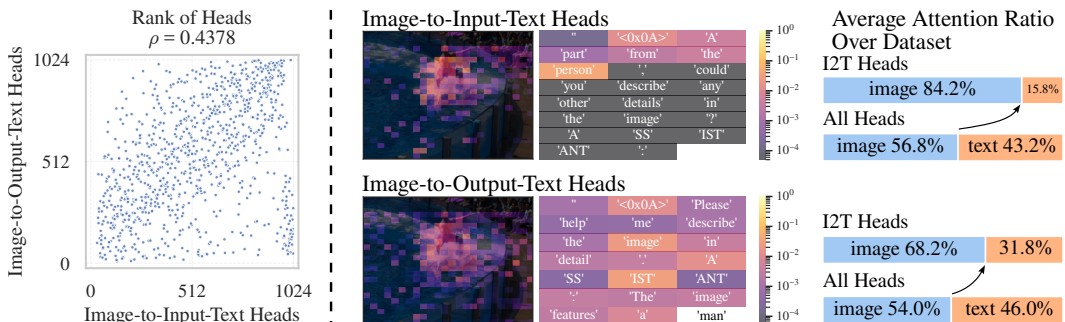

Figure 4: Left: Visualization of the rank distributions of image-to-input-text heads and image-to-output-text heads, showing that the two are largely uncorrelated. Right: Compared to the average of all heads, all image-to-text heads we identified, exhibit a stronger focus on visual content.

coefficient $\rho$ between them. A higher $\rho$ value indicates a greater similarity in the ranking of heads across the datasets being compared.

As shown in Fig. 3, we observe that ***the correlation between text-to-text heads under the same alignment format is very high, but the correlation is significantly lower under different formats.*** Specifically, for Yes/No questions, the correlation coefficient between the heads identified by POPE COCO adversarial and POPE COCO random reaches 0.8247, and the correlation coefficient between the heads identified by POPE COCO adversarial and POPE GQA [22] adversarial reaches 0.8358. However, even with the same images and similar questions, when switching to MCQ format, the correlation coefficient between heads drops to only 0.1233. This demonstrates that our T2T heads exhibit a strong correlation with the alignment format.

**Image-to-input-text causal path is different from image-to-output-text causal path.** For the image-to-text path, we further investigate the behavioral differences between the heads in the image-to-input-text and image-to-output-text pathways. Specifically, we conduct experiments using CHAIR. Unlike the standard CHAIR [45] evaluation, we design a modified prompting strategy: we make certain objects to appear in the input while prompting the LVLM to generate captions for other objects. We then extract the heads separately from the input and output paths and calculated the correlation coefficient $\rho$ between these two sets of heads. Additionally, we calculate the average attention weights of the top-10 identified heads and compared them with the average attention weights of all heads.

As illustrated in Fig. 4, we observe that both image-to-input-text heads and image-to-output-text heads attend to the image more significantly than other heads. ***However, even within the image-to-text causal path, the heads identified in the image-to-input-text and image-to-output-text pathways differ considerably.*** This suggests that focusing solely on the output text when mitigating hallucinations may be insufficient, even though hallucinations only occur in the output.

| Method | POPE (COCO) | | | | | | CHAIR | |
| | *Random* | | *Popular* | | *Advers.* | | | |
| | Acc. | F1 | Acc. | F1 | Acc. | F1 | $C_S\downarrow$ | $C_I\downarrow$ |
|---|---|---|---|---|---|---|---|---|
| w/o mask | 85.1 | 83.7 | 83.7 | 82.4 | 80.9 | 80.0 | 52.2 | 14.6 |
| w/ mask | 83.2 | 81.2 | 82.0 | 80.2 | 79.7 | 78.1 | 62.4 | 32.4 |
| $\Delta\%$ | ↓1.9 | ↓2.5 | ↓1.7 | ↓2.2 | ↓1.2 | ↓1.9 | ↑10.2 | ↑17.8 |

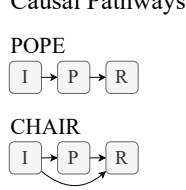

Causal Pathways

Figure 5: Left: When we entirely knocked out the attention weights of output tokens attending to image tokens, POPE's performance worsens very little, whereas CHAIR's performance gets worse significantly. Right: This indicates that LVLMs utilize different pathways for different question formats.

## 3.2 LVLMs adaptively select different causal pathways for different question formats

**Motivation & experiment setting.** Given that both the image-to-input-text-to-output-text and image-to-output-text pathways may contribute to the prediction of final output tokens (*i.e.*, the response), we further investigate whether LVLMs exhibit different pathway utilization patterns across different types of hallucination benchmarks. To demonstrate the significance of both causal pathways in LVLM hallucinations, following [26], we attempt to remove the image-to-text causal path from the image input to the output while preserving the image-to-text causal path from image input to text input and the text-to-text causal path from text input to output. Specifically, we utilize LLaVA [36] to mask the attention from output tokens to image tokens on both POPE and CHAIR, as shown in Fig. 5.

**Finding & discussion.** We observe that on POPE, even when the image input to the output attention is entirely knocked out, performance only decreases by approximately 2%, underscoring the importance of the text-to-text causal path. However, on CHAIR, consistent with the findings of [26], the performance worsens by about 10% after knocking out, indicating the substantial contribution of the image-to-text causal path. This experiment also suggests that *the causal paths taken by LVLM in answering questions differ across different question formats.* Based on these findings, we contend that focusing solely on strengthening one causal path is insufficient for mitigating LVLM hallucinations comprehensively.

## 4 Mitigating hallucinations through multi-path head intervention

In this section, building on the heads identified in our intervention framework (Sec. 2) and their roles in different causal pathways (Sec. 3), we propose a simple yet effective intervention method to reduce hallucinations.

**Important claim**: To demonstrate that our study is general and not specific on questions, **when identifying heads, we use images and questions that are entirely different from those used during testing.** Specifically, for each tested dataset, we generate a new dataset and calculate importance scores on it, ensuring that this newly generated dataset shares only the same format with the original dataset without sharing any images.

To identify attention heads with the most pronounced impact on hallucination, we use Importance Scores $S_{\text{T2T}}^{(l,n)}$ and $S_{\text{I2T}}^{(l,n)}$ as indicators for head selection. Specifically, we select $Z_{\text{T2T}}^{+}$ and $Z_{\text{T2T}}^{-}$, the top and bottom $\xi$ attention heads ranked by alignment scores, which are associated with hallucination suppression and promotion, respectively. Similarly, we select $Z_{\text{I2T}}^{+}$, the top $\zeta$ attention heads ranked by image scores $S_{\text{I2T}}^{(l,n)}$, which exhibit the desirable attention patterns over visual inputs.

Finally, we derive the final sets of attention heads by combining the respective conditions:

$$Z^{-} = Z_{\text{T2T}}^{-}, \qquad Z^{+} = Z_{\text{I2T}}^{+} \cup Z_{\text{T2T}}^{+}$$

To effectively mitigating hallucinations, we apply different scaling factors $\lambda^{(l,n)}$ to different heads according to $Z^{-}$ and $Z^{+}$. Specifically, we apply this modification to MHA, the updated output of the

MHA module can be expressed as follows:

$$\tilde{H}_{\leq t}^{(l)} = \sum_{n=1}^{N} \lambda^{(l,n)} H_{\leq t}^{(l,n)}, \qquad \lambda^{(l,n)} = \begin{cases} \gamma^+, & \text{if } (l,n) \in Z^+ \\ \gamma^-, & \text{if } (l,n) \in Z^- \\ 1, & \text{otherwise} \end{cases}$$

where $\gamma^+$ and $\gamma^-$ are hyperparameters.

## 5 Experiments

### 5.1 Experimental settings

#### 5.1.1 Datasets and metrics

**POPE [29]**, the Polling-based Object Probing Evaluation (POPE) benchmark is a widely utilized benchmark for evaluating object hallucination in LVLMs. Built upon three established datasets, COCO [33], A-OKVQA [46], and GQA [22], POPE samples 500 images from each dataset and selects three ground-truth objects per image as positive instances. Correspondingly, three absent objects are sampled per image using one of three strategies: random, popular, and adversarial sampling. Each object is then queried in the binary format: "*Is there a(n) {object} in the image?*" This yields 3,000 questions per dataset, with a balanced distribution of positive and negative instances.

**MCQ-POPE.** To extend the format of questions, we adapted POPE into multiple-choice questions, creating a new benchmark named MCQ-POPE. Specifically, for each image in each dataset each split, we extract all the three objects labeling Yes and three labeling No. For each object, we pair it with three other objects whose labels differ from the target object, shuffle the four objects, and present them in a multiple-choice question. The resulting MCQ-POPE benchmark also maintains a total of 3,000 questions for each dataset each split, ensuring consistency with the original POPE benchmark.

**CHAIR [45]**, the Caption Hallucination Assessment with Image Relevance is another widely used metric for evaluating hallucinations in LVLMs, focusing on open-ended image captioning tasks. Objects mentioned in the generated captions but absent from the ground truth are regarded as hallucinations. CHAIR evaluates such errors along two dimensions: sentence-level and instance-level, with their respective scores computed as follows:

$$C_S = \frac{|\{\text{captions with hallucinated objects}\}|}{|\{\text{all captions}\}|} \quad \text{and} \quad C_I = \frac{|\{\text{hallucinated objects}\}|}{|\{\text{all mentioned objects}\}|}$$

**MME [17]**, the MLLM Evaluation Benchmark is a benchmark designed to comprehensively evaluate the performance of LVLMs. Similar to POPE, MME also requires the model to answer yes or no questions. Following [74, 27], and given our focus on hallucination, we report results on the hallucination subsets, specifically including existence, count, position, and color subsets. Consistent with their official implementation, we use the sum of accuracy and accuracy+ as the evaluation metric.

**Baselines.** We conduct experiments on the LLaVA-1.5-7B [36]. In addition to the vanilla baseline, we also compared AllPath with some other training-free methods, including contrastive-decoding-based VCD [27] and ICD [61], intervention-based PAI [39] and AD-HH [72].

#### 5.1.2 Implementation details

Unless otherwise specified, we set the scaling factors to $\gamma^+ = 2.0$ and $\gamma^- = 0.0$. For short-answering tasks, we set $\xi = 20$ and $\zeta = 10$, while for open-ended tasks, we set $\xi = 40$ and $\zeta = 50$. For MME, we employ the same attention heads identified from POPE. See Appendix A for more details.

### 5.2 Comparison with state-of-the-art methods

As shown in Table 1, our AllPath exhibits the highest performance across all benchmarks with different formats. It can be observed that on the POPE dataset, our method achieves at least a 2.1% increase in F1, significantly higher than the best baseline improvement of 0.9%. On MCQ-POPE, our method also achieves at least a 5.2% increase. On the CHAIR dataset, we also achieve the highest F1 increase of 2.6%, exceeding the best baseline improvement of 1.9%. Additionally, we find that while

| Method | POPE (COCO) | | | | | | MCQ-POPE (COCO) | | | | | | CHAIR | |
| | *Random* | | *Popular* | | *Advers.* | | *Random* | | *Popular* | | *Advers.* | | | |
| | Acc. | F1 | Acc. | F1 | Acc. | F1 | Acc. | F1 | Acc. | F1 | Acc. | F1 | $C_S\downarrow$ | $C_I\downarrow$ |
|---|---|---|---|---|---|---|---|---|---|---|---|---|---|---|
| Vanilla | 85.1 | 83.7 | 83.7 | 82.4 | 80.9 | 80.0 | 72.8 | 72.9 | 68.0 | 68.0 | 64.4 | 64.5 | 52.2 | 14.6 |
| VCD [27] | 86.3 | 85.2 | 84.5 | 83.5 | 81.4 | 80.9 | 78.2 | 78.2 | 72.2 | 72.2 | 67.9 | 67.9 | 58.2 | 16.1 |
| ICD [61] | 84.4 | 82.2 | 83.8 | 81.6 | 81.5 | 79.5 | 69.1 | 69.1 | 67.2 | 67.2 | 64.7 | 64.6 | 51.4 | 14.7 |
| PAI [39] | 86.4 | 85.0 | 84.9 | 83.6 | 82.5 | 81.4 | 78.0 | 78.0 | 71.2 | 71.2 | 68.0 | 67.9 | 28.8 | 7.9 |
| AD-HH [72] | 85.0 | 83.6 | 83.7 | 82.4 | 80.9 | 79.9 | 78.5 | 78.6 | 71.1 | 71.1 | 68.1 | 68.1 | 33.2 | 7.5 |
| **AllPath** | **87.2** | **86.0** | **86.0** | **84.9** | **82.8** | **82.1** | **80.5** | **80.5** | **74.0** | **74.0** | **69.7** | **69.7** | **26.6** | **7.2** |

Table 1: Results on POPE, MCQ-POPE, and CHAIR benchmark. The F1 score for MCQ-POPE represents Macro F1.

| Method | Object-Level | | Attribute-Level | | Total↑ |
| | *Existence*↑ | *Count*↑ | *Position*↑ | *Color*↑ | |
|---|---|---|---|---|---|
| Vanilla | 180.0 (±8.66) | 113.9 (±10.05) | 116.7 (±15.90) | 129.4 (±18.28) | 540.0 (±39.69) |
| VCD [27] | 177.8 (±2.55) | 122.8 (±20.84) | 122.2 (±1.92) | 141.7 (±2.89) | 564.4 (±20.84) |
| ICD [61] | 185.0 (±5.00) | 121.7 (±2.89) | 118.3 (±6.01) | 151.7 (±7.64) | 576.7 (±11.67) |
| PAI [39] | 185.0 (±5.00) | 122.8 (±14.94) | 114.4 (±4.19) | 144.4 (±5.09) | 566.7 (±15.90) |
| AD-HH [72] | 179.4 (±5.09) | 117.8 (±12.51) | 116.1 (±13.37) | 152.8 (±10.72) | 566.1 (±22.19) |
| **AllPath** | **188.3** (±2.89) | **126.1** (±0.96) | **132.2** (±8.39) | **153.3** (±7.64) | **600.0** (±8.66) |

Table 2: Performance on MME Benchmark [17]. Results are reported as the average of 3 test runs, with ± indicating the sample standard deviation.

VCD shows noticeable improvements on POPE and MCQ-POPE, its performance on CHAIR suffers a significant decline of $1.4\%$. AD-HH, on the other hand, exhibits substantial performance gains on CHAIR and MCQ-POPE, but its performance on POPE is similar to that of the vanilla method. Our method achieves substantial performance gains across three benchmarks of completely different formats.

Our results on MME is shown in Table 2. It can be observed that our method still achieves stable performance improvement on the MME hallucination subset. Considering that we still use the POPE heads for MME without extracting independent heads specifically for MME, this demonstrating the generalizability of our method under a unified task format.

## 5.3 Ablation study

**Ablation study on pathways.** To evaluate the two key components of our method, the T2T heads and I2T heads, we conduct an ablation study on POPE. The results are shown in the upper part of Table 3. We observe that employing either T2T heads or I2T heads individually leads to a certain degree of performance improvement. However, the most substantial performance gain is achieved only when all heads are utilized, significantly outperforming the previous two configurations. This result substantiates our claim that both proposed pathways are present and exert a considerable influence on LVLM hallucinations. Therefore, considering only one pathway is insufficient for effectively mitigating LVLM hallucinations.

**Ablation study on hyperparameters.** To validate that our AllPath is insensitive to hyperparameter selection, we perform an ablation study on POPE COCO. As results report in the lower part of Table 3, our method is largely insensitive to the choice of hyperparameters. Substantial changes to the hyperparameters lead to only minor variations in performance. For instance, the accuracy on the random subset fluctuates only within the range of 85.6–88.3, consistently outperforming the baseline of 85.1. Notably, some hyperparameter settings in this table result in better performance on POPE. This is because rather than tuning hyperparameters for a specific dataset, we select a single set of hyperparameters that works best across various discriminative tasks. This shows that our method is effective and stable without extensive tuning.

| Hyperparameters | | | | Random | | Popular | | Adversarial | |
|---|---|---|---|---|---|---|---|---|---|
| $\xi$ | $\zeta$ | $\gamma^+$ | $\gamma^-$ | Acc. | F1 | Acc. | F1 | Acc. | F1 |
| 20 | 10 | 2.0 | 0.0 | 87.2 | 86.0 | 86.0 | 84.9 | 82.8 | 82.1 |
| **0** | 10 | 2.0 | 0.0 | 86.2 | 85.2 | 84.5 | 83.6 | 81.6 | 81.1 |
| 20 | **0** | 2.0 | 0.0 | 86.4 | 84.9 | 85.2 | 83.7 | 82.5 | 81.3 |
| **0** | **0** | 2.0 | 0.0 | 85.1 | 83.7 | 83.7 | 82.4 | 80.9 | 80.0 |
| **10** | 10 | 2.0 | 0.0 | 87.3 | 86.2 | 86.0 | 85.0 | 82.6 | 82.0 |
| **30** | 10 | 2.0 | 0.0 | 88.3 | 87.7 | 86.7 | 86.3 | 82.9 | 83.0 |
| **40** | 10 | 2.0 | 0.0 | 88.1 | 87.9 | 85.9 | 86.0 | 81.2 | 82.2 |
| 20 | **5** | 2.0 | 0.0 | 86.2 | 84.8 | 85.8 | 84.6 | 82.7 | 81.9 |
| 20 | **15** | 2.0 | 0.0 | 88.2 | 87.4 | 86.6 | 85.9 | 83.0 | 82.8 |
| 20 | **20** | 2.0 | 0.0 | 88.1 | 87.3 | 86.7 | 86.0 | 83.0 | 82.8 |
| 20 | 10 | **1.2** | 0.0 | 85.6 | 85.3 | 85.3 | 84.1 | 82.3 | 81.5 |
| 20 | 10 | **1.5** | 0.0 | 86.9 | 85.7 | 85.6 | 84.5 | 82.5 | 81.7 |
| 20 | 10 | **2.5** | 0.0 | 86.8 | 85.6 | 86.3 | 85.4 | 83.1 | 82.5 |
| 20 | 10 | 2.0 | **0.2** | 87.1 | 85.9 | 85.8 | 84.7 | 82.5 | 81.8 |
| 20 | 10 | 2.0 | **0.5** | 87.0 | 85.8 | 85.6 | 84.5 | 82.2 | 81.5 |
| 20 | 10 | 2.0 | **0.8** | 86.8 | 85.5 | 85.4 | 84.2 | 82.0 | 81.3 |

Table 3: Ablation studies. Hyperparameters different from the default setting are highlighted.

# 6 Related work

**Large Vision-Language Models.** With the success of LLMs [1, 59, 60, 10, 44, 2], an increasing number of studies focus on LVLMs [36, 3, 86, 8, 9, 73]. A typical LVLM architecture comprises a vision encoder [43] with an LLM, connected via a modality alignment module such as a linear projection [36] or a Q-former [86, 14]. To support effective cross-modal reasoning, these models follow a multi-stage training pipeline. The pretraining stage focuses on aligning visual and language modalities using large-scale image-text pairs. Building on this foundation, supervised fine-tuning (SFT) with instruction-following data enhances the model's ability to generate coherent responses conditioned on multimodal prompts. Beyond standard training recipe, several works have further explored advanced adaptation strategies, including multi-phase pre-training [4], knowledge learning stage [37], and Reinforcement Learning from Human Feedback [55].

**Mitigating hallucinations in LVLMs.** Extensive researches have been conducted to mitigate LVLM hallucinations from various perspectives, including curating training datasets of models [35, 75, 79], modifying training objectives [24], adjusting model architectures [65], and applying post-processing methods to model outputs [85, 63]. Nevertheless, many studies also focus on mitigating hallucinations during inference. Some work [16, 27, 62, 61, 82] primarily focused on Contrastive Decoding [28] methods, while more recent approaches have focused on attention mechanisms, such as applying straightforward scaling techniques [39] or by dynamically adjusting scaling weights [25] on attention scores for a range of layers and tokens. Meanwhile, some approaches focus on modifying attention heads. [72] identifies hallucinatory heads by zero-ablation, and [7] identifies hallucinatory heads by modifying input image tokens and training a binary classifier.

**Understanding LVLMs.** Currently, methods for understanding LVLMs can mainly be categorized into three types. The first type is based on averaging transformer weights. However, [54, 72] demonstrates that they do not effectively capture the specific regions that the model actually attends to. Additionally, Grad-CAM [47] visualize the model's focus by calculating the gradient of the output with respect to the input. Another approach is the early exit method [42, 72], which obtains hidden state representations by extracting them before the unembedding process.

# 7 Conclusion

In this paper, we propose a unified framework for mitigating hallucinations in LVLMs. By distinguishing between Text-to-Text and Image-to-Text pathways and analyzing their corresponding attention heads, we develop AllPath, a simple yet powerful method. Experimental results demonstrate that AllPath consistently achieves superior performance across all benchmarks.

**Acknowledgment:** This work is supported by the National Natural Science Foundation of China under Grant No.62206174 and No.62576365.

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

# Appendix

The appendix includes supplementary details, analyses, and experiments in support of the main text, structured as follows:

## A    Additional experimental and implementation details

### A.1    Additional implementation details

For POPE [29] and MCQ-POPE, attention heads are identified using samples generated via an adversarial strategy, and subsequently applied across all three test splits: random, popular, and adversarial. Following the official LLaVA evaluation setup, prompts are appended with "Please answer this question with one word." All evaluations on POPE, MCQ-POPE, and MME [17] utilized nucleus sampling with a temperature of 1, and does not use beam search.

For CHAIR [45], we randomly sample 500 images from the COCO [33] dataset validation subset to identify attention heads. For both attention head identification and final evaluation, we pair each image with the prompt "*Please help me describe the image in detail.*", and responses are generated using greedy decoding with a maximum length of 512 tokens.

### A.2    Prompt for MCQ-POPE

> Which of the following {appears | does not appear} in the image?
> A. {object1}
> B. {object2}
> C. {object3}
> D. {object4}

## B    Additional analysis

In this section, we first supplement the main text with additional analyses based on LLaVA-v1.5-7B [36] in Section B.1 to further support the conclusion that LVLMs adaptively select different causal pathways for different questions. We then replicate the same series of analyses using Qwen-VL-Chat [3] in Section B.2 and Section B.3. Implementation details for Qwen-VL-Chat are provided in Section C.1.

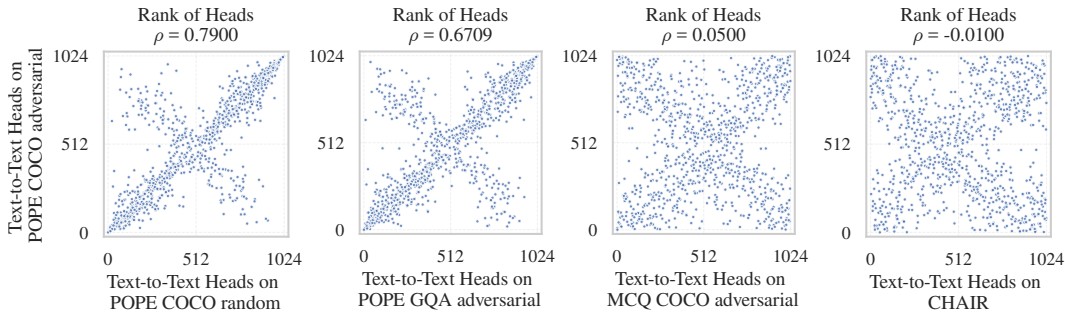

Figure 6: Visualization of the rank distributions of text-to-text heads extracted from different datasets on Qwen-VL-Chat [3]. $\rho$ denotes the correlation coefficient; a higher value (*i.e.*, points concentrated along the diagonal) indicates greater similarity between the two sets of heads. Heads identified from datasets with similar question–answer alignment formats exhibit strong correlation and substantial overlap.

| Method | LLaVA on POPE (COCO) | | | | | | Qwen on POPE (COCO) | | | | | |
| | *Random* | | *Popular* | | *Advers.* | | *Random* | | *Popular* | | *Advers.* | |
| | Acc. | F1 | Acc. | F1 | Acc. | F1 | Acc. | F1 | Acc. | F1 | Acc. | F1 |
|---|---|---|---|---|---|---|---|---|---|---|---|---|
| w/o mask | 85.1 | 83.7 | 83.7 | 82.4 | 80.9 | 80.0 | 84.5 | 82.3 | 83.4 | 81.4 | 80.5 | 78.9 |
| w/ mask | 52.9 | 59.2 | 47.8 | 56.7 | 48.9 | 56.6 | 48.3 | 31.0 | 51.2 | 31.3 | 50.6 | 33.5 |
| $\Delta\%$ | ↓32.2 | ↓24.5 | ↓35.9 | ↓25.7 | ↓32.0 | ↓23.4 | ↓35.0 | ↓51.3 | ↓32.2 | ↓50.1 | ↓29.9 | ↓45.4 |

Table 4: When we knock out the attention weights of ***input text*** tokens attending to image tokens on LLaVA-v1.5-7b [36] and Qwen-VL-Chat [3], POPE's performance worsens significantly.

### B.1 Adaptive selection of causal path in LLaVA-v1.5-7B

As a supplement to the main experiments, we further knock out the image-to-text causal path from the image input to the text input while preserving the image-to-text path from image input to text output, as well as the text-to-text path from text input to output. Specifically, our experiments are conducted on LLaVA, where we mask the attention from text input tokens to image tokens.

As shown in Table 4 (left), this modification results in a substantial performance drop on POPE, with accuracy falling to **around 50%, comparable to random guessing**. This suggests that the model's prediction on POPE heavily depends on the causal path from the image input to the text input.

### B.2 Functional interpretation of the identified heads in Qwen-VL-Chat

**Text-to-text causal path is related to the alignment.** We present the correlation maps of heads identified in Qwen-VL-Chat in Fig. 6. Consistent with the findings from LLaVA, we can observe that *the correlation between text-to-text heads under the same alignment format is very high, but the correlation is significantly lower under different formats* in Qwen-VL-Chat. Specifically, for Yes/No questions, the heads identified from POPE COCO adversarial show strong correlations with those from POPE COCO random and POPE GQA [22] adversarial, with correlation coefficients of 0.79 and 0.67, respectively. However, the correlation drops markedly when comparing POPE COCO adversarial with heads identified from the MCQ-POPE COCO (0.05), and also declines when compared to those from generation tasks (-0.01). These observations indicate that the T2T heads in Qwen-VL-Chat also exhibit a strong correlation with the alignment format.

**Image-to-input-text causal path is different from image-to-input-text causal path.** Consistent with the LLaVA-based analysis in the main text, Fig. 7 presents the correlation between image-to-input-text and image-to-output-text heads in Qwen-VL-Chat under the modified prompting strategy on CHAIR, along with the visualization of their focus on visual content. The results reveal that Qwen-VL-Chat exhibits a similar pattern: although both the image-to-input-text and image-to-output-text heads attend more strongly to the image than other heads, their attention patterns differ markedly. This suggests that the two pathways modulate the model's behavior in distinct.

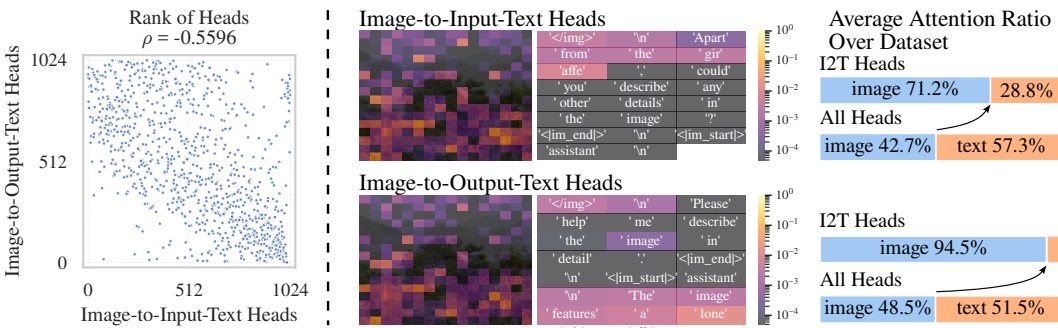

Figure 7: Left: Visualization of the rank distributions of image-to-input-text heads and image-to-output-text heads on Qwen-VL-Chat [3], showing that the two are largely uncorrelated. Right: Compared to the average of all heads, all image-to-text heads we identified, exhibit a stronger focus on visual content on Qwen-VL-Chat.

| Method | POPE (COCO) | | | | | | CHAIR | | Causal Pathways |
|---|---|---|---|---|---|---|---|---|---|
| | Random | | Popular | | Advers. | | $C_S\downarrow$ | $C_I\downarrow$ | |
| | Acc. | F1 | Acc. | F1 | Acc. | F1 | | | |
| w/o mask | 84.5 | 82.3 | 83.4 | 81.4 | 80.5 | 78.9 | 43.4 | 13.5 | |
| w/ mask | 83.4 | 81.1 | 83.1 | 80.9 | 80.9 | 78.9 | 78.4 | 48.0 | |
| $\Delta\%$ | ↓1.1 | ↓1.2 | ↓0.3 | ↓0.5 | ↑0.4 | ↓0.0 | ↑35.0 | ↑34.5 | |

Causal Pathways

POPE
$I \to P \to R$

CHAIR
$I \to P \to R$

Figure 8: Left: When we mask the attention weights of **output** tokens attending to image tokens on Qwen-VL-Chat [3], POPE's performance worsens very little, whereas CHAIR's performance gets worse significantly. Right: This indicates that LVLMs utilize different pathways for different question formats.

## B.3 Adaptive selection of causal path in Qwen-VL-Chat

We perform two counterfactual interventions based on Qwen-VL-Chat: (1) removing the image-to-output-text path by masking attention from text output tokens to image tokens, while preserving the image-to-input-text and text-to-text paths; and (2) removing the image-to-input-text path by masking attention from text input tokens to image tokens, while keeping the image-to-output-text and text-to-text paths. The effects of both interventions are illustrated in Fig. 8 and Table 4 (right).

(1) As shown in the left panel of Fig. 8, the first counterfactual intervention results in a negligible impact on POPE, with an average performance drop of less than 0.5 points, whereas it causes a substantial degradation on CHAIR. This suggests that the image-to-output path is not critical for POPE, but plays a crucial role in CHAIR. (2) Table 4 provides further evidence of the critical role played by the image-to-input-text causal path in POPE: masking this pathway reduces the model's accuracy to chance level, underscoring its essential contribution to successful prediction.

## C Additional experimental results

### C.1 Experimental setup for Qwen-VL-Chat

Following a similar methodology to LLaVA-v1.5-7B [36], we identify attention heads in Qwen-VL-Chat [3] using a generated dataset that is entirely disjoint from the test set. This dataset is employed to compute both the T2T and I2T Scores for head probing. However, due to substantial differences between the base models of Qwen-VL-Chat and LLaVA, we introduce two key adaptations in our implementation: (1) We modify the I2T Score to $S_{\text{I2T}}^{(l,n)} = S_{\text{I2T}}^{(l,n),+}$. This change is necessitated by the observation that attention heads in Qwen-VL-Chat generally assign non-negligible attention to most image patches. As a result, penalizing background attention would otherwise favor heads with uniformly low attention to visual inputs, which is not desirable. (2) We adopt $\xi = 20$ and $\zeta = 10$ for both discriminative and generative tasks.

| Method | POPE (COCO) | | | | | | MCQ-POPE (COCO) | | | | | | CHAIR | | | |
| | Random | | Popular | | Advers. | | Random | | Popular | | Advers. | | $C_S\downarrow$ | $C_I\downarrow$ | F1 | Len |
| | Acc. | F1 | Acc. | F1 | Acc. | F1 | Acc. | F1 | Acc. | F1 | Acc. | F1 | | | | |
|---|---|---|---|---|---|---|---|---|---|---|---|---|---|---|---|---|
| Vanilla | 84.5 | 82.3 | 83.4 | 81.4 | 80.5 | 78.9 | 61.8 | 67.3 | 55.9 | 58.9 | 52.5 | 56.1 | 43.4 | 13.5 | 76.4 | 98.1 |
| VCD [27] | 85.5 | 83.4 | 84.3 | 82.3 | 82.0 | 80.2 | 66.6 | 71.9 | 60.9 | 63.6 | 55.3 | 58.3 | 46.0 | 12.9 | 76.9 | 100.8 |
| ICD [61] | 85.4 | 83.6 | 84.1 | 82.4 | 81.7 | 80.3 | 74.9 | **77.7** | 67.3 | **68.8** | 62.0 | 63.3 | 50.4 | 14.4 | 73.9 | 100.5 |
| PAI [39] | 85.8 | 84.1 | 84.9 | 83.3 | **82.4** | 80.9 | 57.9 | 66.1 | 52.0 | 56.6 | 48.6 | 54.6 | 17.2 | 7.9 | 70.1 | 36.8 |
| **AllPath** | **86.6** | **85.3** | **85.6** | **84.3** | **82.4** | **81.5** | **75.4** | **77.7** | **68.8** | **68.8** | **62.6** | **63.6** | **34.2** | **9.0** | **77.2** | 91.0 |

Table 5: Results of Qwen-VL-Chat [3] model on POPE [29], MCQ-POPE, and CHAIR [45] benchmark. The F1 score for MCQ-POPE represents Macro F1. Notably, PAI compromises generative capability, resulting in a substantial reduction in generation length. In contrast, our method significantly improves performance across all benchmarks while largely maintaining generative capacity.

| Method | POPE (A-OKVQA) | | | | | | POPE (GQA) | | | | | |
| | Random | | Popular | | Advers. | | Random | | Popular | | Advers. | |
| | Acc. | F1 | Acc. | F1 | Acc. | F1 | Acc. | F1 | Acc. | F1 | Acc. | F1 |
|---|---|---|---|---|---|---|---|---|---|---|---|---|
| Vanilla | 86.3 | 85.8 | 82.7 | 82.7 | 75.9 | 77.5 | 85.1 | 84.5 | 79.6 | 80.0 | 76.6 | 77.6 |
| VCD [27] | 86.2 | 86.2 | 82.3 | 83.0 | 76.2 | 78.5 | 86.4 | 86.4 | 78.3 | 79.8 | 75.6 | 77.7 |
| ICD [61] | 86.0 | 85.1 | 83.2 | 82.6 | 77.3 | 77.8 | 84.5 | 83.3 | 80.6 | 80.0 | 77.5 | 77.6 |
| PAI [39] | 87.8 | 87.3 | 84.1 | 84.0 | 77.2 | 78.5 | 86.2 | 85.6 | 80.5 | 80.8 | 77.5 | 78.4 |
| AD-HH [72] | 86.2 | 85.7 | 82.7 | 82.7 | 76.0 | 77.6 | 85.0 | 84.5 | 79.6 | 80.0 | 76.5 | 77.5 |
| **AllPath** | **88.1** | **87.9** | **84.2** | **84.5** | **77.6** | **79.4** | **87.5** | **87.2** | **81.9** | **82.5** | **78.2** | **79.5** |

Table 6: Results of LLaVA-v1.5-7b [36] model on POPE [29] A-OKVQA [46] and GQA [22] split. Our method achieves consistent improvements over previous approaches across all metrics.

## C.2 Results of Qwen-VL-Chat on POPE, MCQ-POPE and CHAIR

The results of Qwen-VL-Chat on POPE [29], MCQ-POPE, CHAIR [45] are shown in Table 5. Notably, our method outperforms each method on at least one benchmark by a significant margin. Specifically, compared to VCD [27], our method achieves an average improvement of **8.0%** and **5.4%** in terms of accuracy and F1 score on MCQ-POPE, respectively, and yields a gain of **11.8** and **3.9** on CHAIR$_S$ and CHAIR$_I$. Relative to ICD [61], our approach demonstrates a significant margin of **16.2** on CHAIR$_S$ and **5.4** on CHAIR$_I$. As for the PAI method [39], it is noteworthy that it degrades the generative capability of Qwen-VL-Chat, leading to a substantial reduction (from 98.1 to 36.8) in generation length on CHAIR. In contrast, our method not only achieves a significant average improvement of **16.1%** on accuracy and **10.9%** on F1 score on MCQ-POPE over PAI, but also substantially reduces the metrics on CHAIR$_S$ and CHAIR$_I$ while largely preserving the original generation length of Qwen-VL-Chat.

## C.3 Results of Qwen2.5-VL on POPE

To verify that our AllPath remains effective on newer and larger models, we evaluate its performance on POPE on Qwen2.5-VL-7B/72B [4], as shown in Table 7. It is worth noting that since 72B version of Qwen2.5-VL has 5 times as many heads as LLaVA, we accordingly scale the number of heads used by a factor of five, *i.e.*, $\xi = 100$ and $\zeta = 50$. As can be seen from the table, our method also achieves consistent improvements on Qwen2.5.

# D Qualitative results

## D.1 Qualitative Results on POPE

Fig. 9 and 10 each illustrate two qualitative results from POPE [29], where the ground-truth answers to the given questions are Yes and No, respectively. They also show attention distributions of top- and bottom-ranked heads by T2T Score during answer generation, and I2T Scores at the object token, with average attention for comparison. From these visualizations, we observe that heads with the highest T2T Scores consistently attend to object tokens, differing from the average pattern. Moreover,

| Method | Qwen2.5-VL-7B / POPE (COCO) | | | | | | Qwen2.5-VL-72B / POPE (COCO) | | | | | |
|---|---|---|---|---|---|---|---|---|---|---|---|---|
| | *Random* | | *Popular* | | *Advers.* | | *Random* | | *Popular* | | *Advers.* | |
| | Acc. | F1 | Acc. | F1 | Acc. | F1 | Acc. | F1 | Acc. | F1 | Acc. | F1 |
| Vanilla | 87.6 | 86.0 | 86.6 | 85.1 | 85.5 | 84.0 | 87.7 | 86.2 | 86.6 | 85.1 | 85.6 | 84.1 |
| **AllPath** | **90.6** | **90.0** | **88.6** | **88.1** | **86.6** | **86.3** | **90.7** | **90.2** | **88.5** | **88.1** | **85.7** | **85.7** |

Table 7: Results of Qwen2.5-VL [4] model on POPE [29] COCO [33] split. Our method achieves consistent improvements across all metrics.

when the ground-truth answer is Yes, *i.e.*, the queried object indeed exists in the image, the heads with the highest I2T Scores focus on the corresponding visual regions, again showing a distinct difference from the mean attention. These qualitative findings further demonstrate that our two core metrics, the I2T Score and T2T Score can effectively identify the critical attention heads along the two pathways. This provides strong evidence that our method, AllPath, mitigates hallucinations by enhancing both pathways.

### D.2 Qualitative Results on CHAIR

To further illustrate the behavior of our intervention, we present qualitative examples highlighting both successful (Fig. 11) and failure (Fig. 12) cases of our AllPath on CHAIR [45]. We emphasize that our method is particularly effective when the LVLM demonstrates correct reasoning even along just one of the two pathways (e.g., image-to-text or text-to-text). In such scenarios, our intervention adaptively enhances the pathway to make it more reliable, thereby reducing hallucination. However, our method is less effective when both pathways in the LVLM exhibit substantial errors, making it difficult for the intervention to fully correct the output.

- **Success Case.** The baseline model generates a hallucination "potted plants", whereas our AllPath correctly describes the scene without hallucination. Visualization shows that when baseline is generating "potted plants", the heads with top I2T Scores correctly attend to the flower in the image, while the heads with bottom T2T scores distribute attention across general objects such as "dog", "bench", and "fence". Our method adaptively amplified both the image-to-text and text-to-text pathways, reinforcing useful information and effectively suppressing hallucination.
- **Failure Case.** The baseline response hallucinates two "bowls", and our AllPath still produces a hallucinated "cup". Visualization shows that when baseline is generating "bowls", the heads with top I2T Score fail to focus on the relevant area in the image, while the heads with bottom T2T Score remain strongly influenced by textual priors, particularly the token "blender". Since neither pathway is reliable, our intervention could not mitigate this hallucination.

## E Limitations

This work focuses on understanding how hallucinations in LVLMs relate to their internal causal pathways. However, we do not investigate what causal pathways are most effective for different types of questions, nor do we explore training strategies that could systematically improve the use of such pathways. Additionally, our analysis is constrained by the scope of existing hallucination benchmarks, which may not capture a broader range of unrecognized or subtle hallucinations. These remain important open questions for future research.

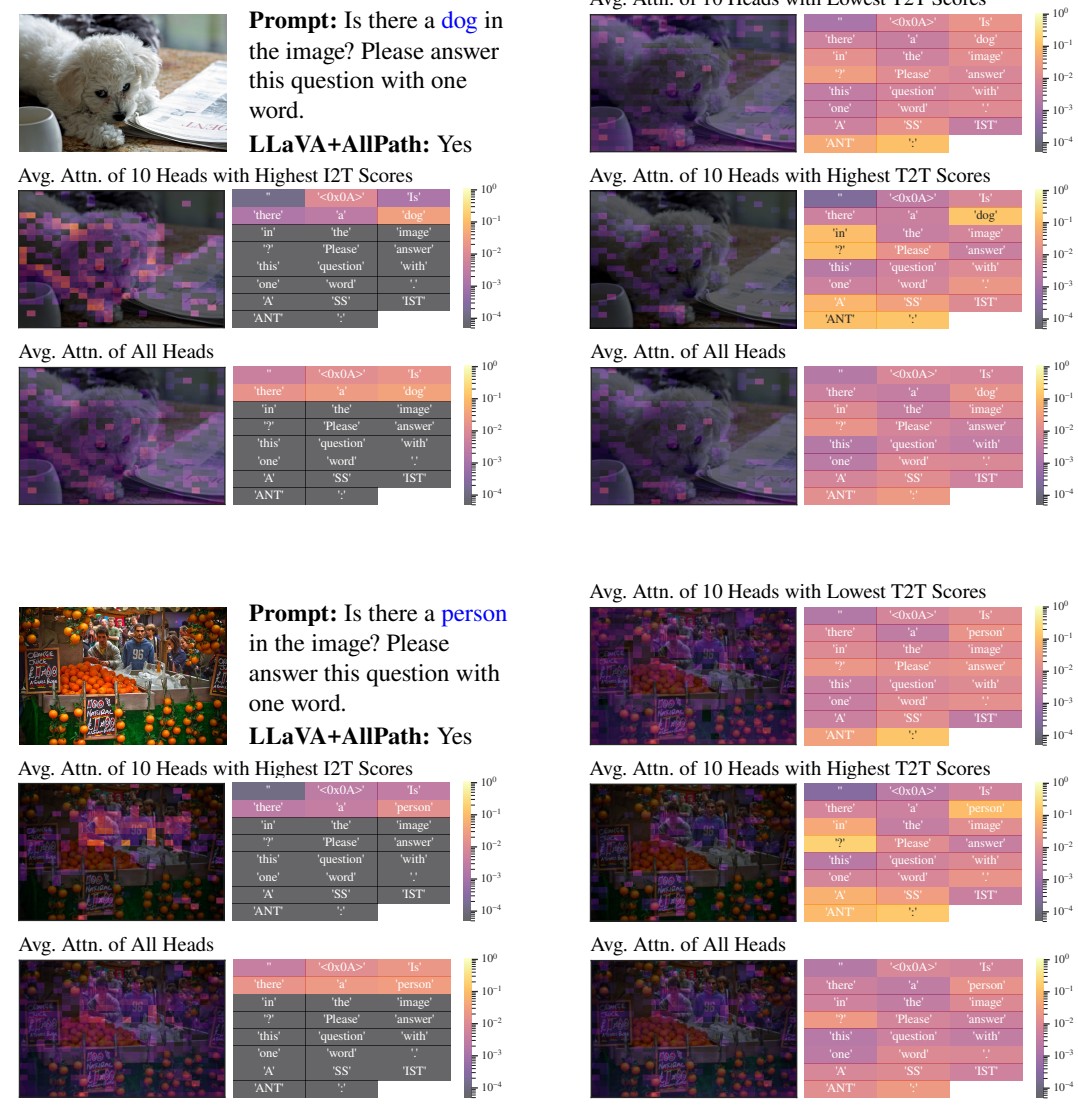

Figure 9: Qualitative visualization of attention weights of I2T and T2T heads on the POPE dataset [29]. Each case contains three rows and two columns. Left: The top row shows the input image, the corresponding question, and the model's predicted answer, where the object token in the question is highlighted in blue. The second and third rows visualize the attention from the object token to previous tokens: the second row shows the average attention of the 10 heads with the highest I2T scores, and the third row shows the average attention of all heads. Right: The attention weights when generating the final answer token. From top to bottom: average attention of the 10 heads with the lowest T2T scores, average attention of the 10 heads with the highest T2T scores, and average attention of all heads.

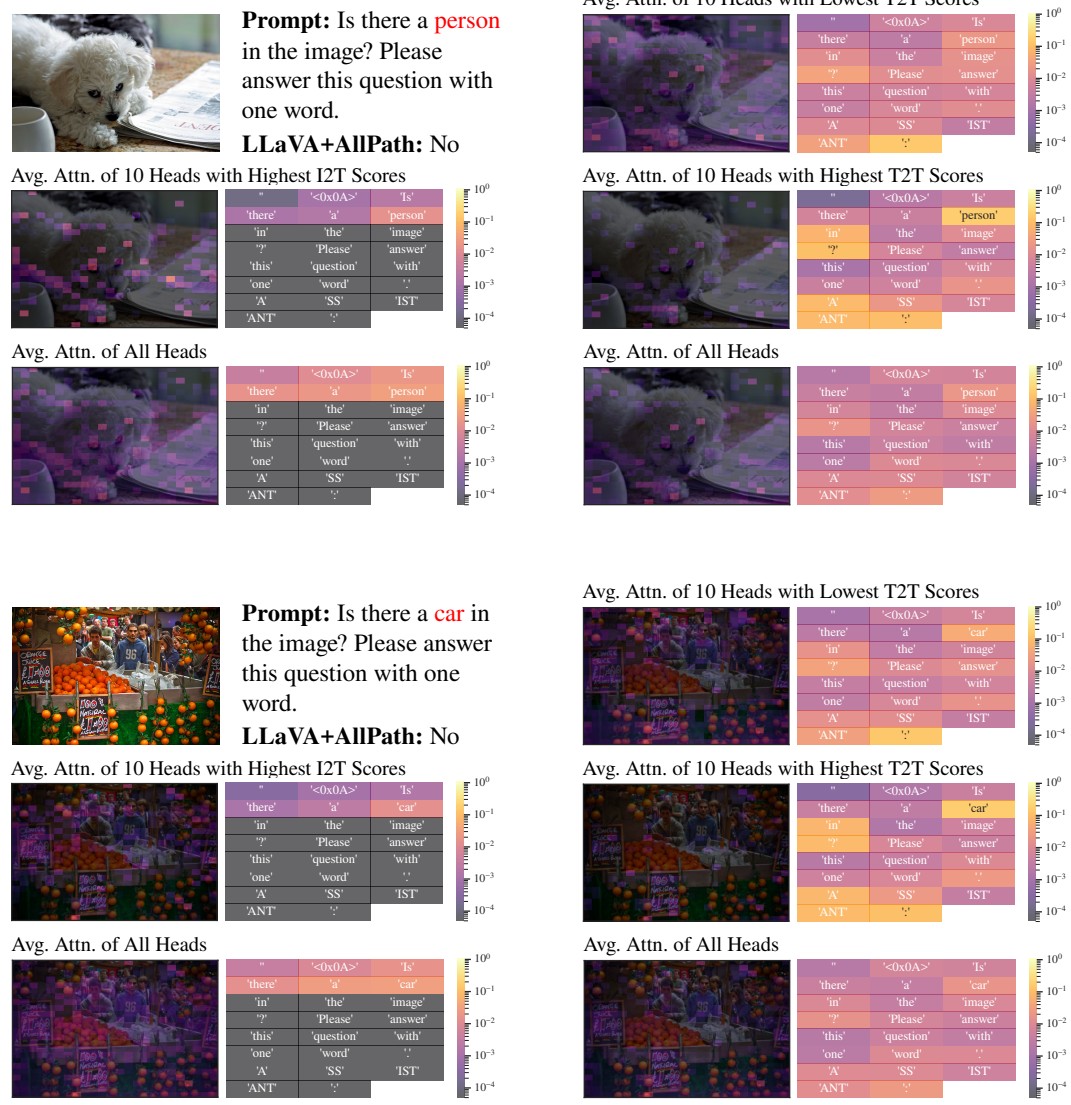

Figure 10: Qualitative visualization of attention weights of I2T and T2T heads on the POPE dataset [29]. Each case contains three rows and two columns. Left: The top row shows the input image, the corresponding question, and the model's predicted answer, where the object token in the question is highlighted in red. The second and third rows visualize the attention from the object token to previous tokens: the second row shows the average attention of the 10 heads with the highest I2T scores, and the third row shows the average attention of all heads. Right: The attention weights when generating the final answer token. From top to bottom: average attention of the 10 heads with the lowest T2T scores, average attention of the 10 heads with the highest T2T scores, and average attention of all heads.

**Prompt:**

Please help me describe the image in detail.

**LLaVA (Baseline):**

The image features a dog sitting on a bench, possibly a park bench, with its tongue out. The dog is wearing a red leash, and it appears to be enjoying its time outdoors. The bench is located near a fence, and there are several potted plants surrounding the area.

In addition to the main dog, there are two other dogs in the scene, one located near the left edge of the image and the other towards the right side. The presence of multiple dogs suggests that this might be a popular spot for pet owners to bring their pets for a walk or a rest.

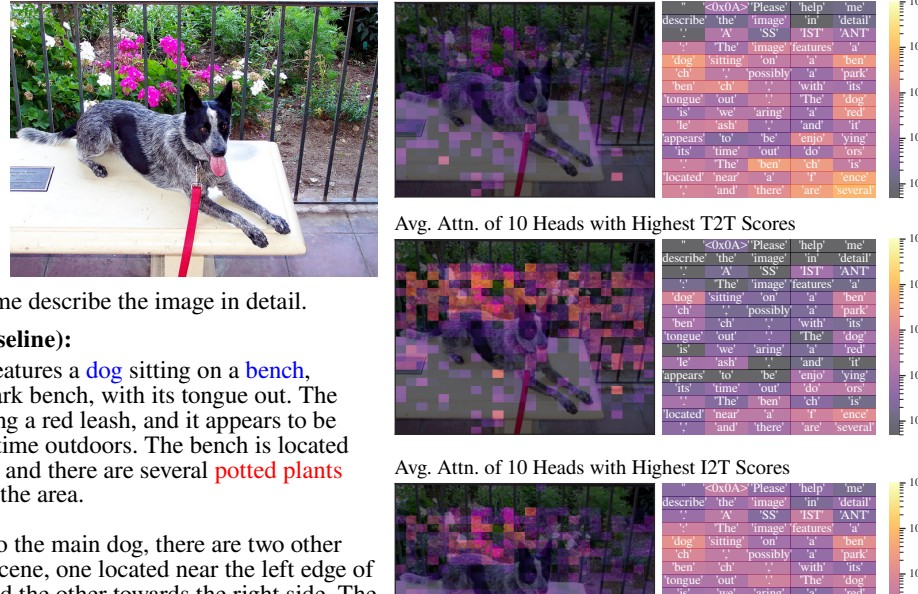

Avg. Attn. of 10 Heads with Lowest T2T Scores

Avg. Attn. of 10 Heads with Highest T2T Scores

Avg. Attn. of 10 Heads with Highest I2T Scores

------------------------------------------------------------

**LLaVA+AllPath:**

The image features a dog sitting on a bench, possibly a park bench, with its tongue out. The dog is wearing a red leash, and it appears to be enjoying its time outdoors. The bench is located near a fence, and there are several potted plants surrounding the area.

In addition to the main dog, there are two other dogs in the scene, one located near the left edge of the image and the other towards the right side. The presence of multiple dogs suggests that this might be a popular spot for pet owners to bring their pets for a walk or a rest.

Figure 11: Qualitative success case of our intervention on CHAIR [45]. Non-hallucinated objects in model's response are highlighted in blue, while hallucinated objects are highlighted in red. The upper part of this figure presents the baseline output, with right column visualizes the model's average attention of different set of heads when generating the hallucinated object "potted plants". From top to bottom: average attention of the 10 heads with the lowest T2T scores, average attention of the 10 heads with the highest T2T scores, and average attention of the 10 heads with the highest I2T scores.

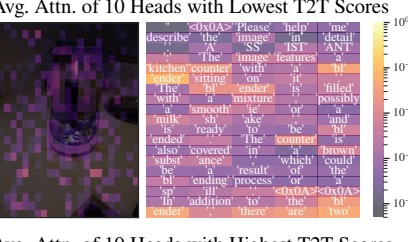

Avg. Attn. of 10 Heads with Lowest T2T Scores

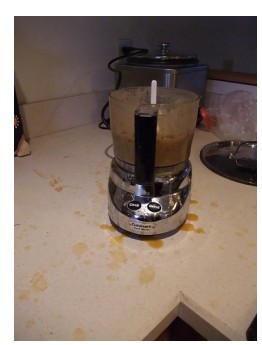

**Prompt:**

Please help me describe the image in detail.

**LLaVA (Baseline):**

The image features a kitchen counter with a blender sitting on it. The blender is filled with a mixture, possibly a smoothie or a milkshake, and is ready to be blended. The counter is also covered in a brown substance, which could be a result of the blending process or a spill.

In addition to the blender, there are two bowls on the counter. One bowl is located on the left side of the blender, while the other is positioned on the right side. The kitchen counter appears to be a bit messy, with the blender and the brown substance taking up a significant portion of the space.

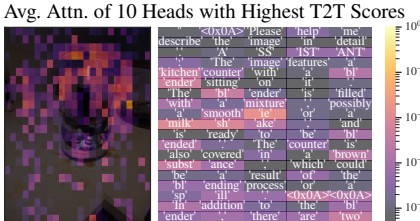

Avg. Attn. of 10 Heads with Highest T2T Scores

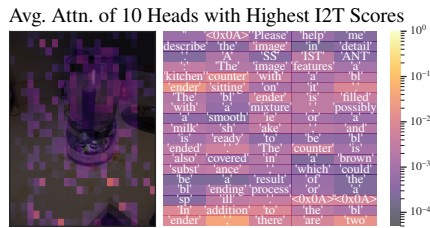

Avg. Attn. of 10 Heads with Highest I2T Scores

---

**LLaVA+AllPath:**

The image features a kitchen counter with a blender sitting on it. The blender is filled with a mixture, possibly a smoothie or a smoothie-like substance. The blender is turned on, and the blender cup is filled with the mixture. The counter is also covered in a bit of mess, with a few drops of liquid spread across the counter.

Figure 12: Qualitative failure case of our intervention on CHAIR [45]. Non-hallucinated objects in model's response are highlighted in blue, while hallucinated objects are highlighted in red. The upper part of this figure presents the baseline output, with right column visualizes the model's average attention of different set of heads when generating the hallucinated object "bowls". From top to bottom: average attention of the 10 heads with the lowest T2T scores, average attention of the 10 heads with the highest T2T scores, and average attention of the 10 heads with the highest I2T scores.

