# OpenReview forum: "Intervene-All-Paths: Unified Mitigation of LVLM Hallucinations across Alignment Formats"
_NeurIPS.cc/2025/Conference — NeurIPS 2025 poster_

### Official Review · Reviewer_5FfF · 2025-06-16

**Clarity:** 2
**Significance:** 3
**Originality:** 3
**Rating:** 4
**Confidence:** 2

**Summary:**

The authors found that the hallucinations in LVLMs do not originate from a single path, but are the result of the interaction of multiple paths, image-to-input-text, image-to-output-text, and text-to-text. The authors also found that LVLM will choose different paths according to the homogeneous format of "question-answer". Based on this, this article proposes Intervene-All-Paths, a method to identify and intervene in the key attention heads that drive hallucinations in each path, and to design intervention strategies for both discriminative and generative formats.

**Questions:**

1. Some hyperparameters mentioned in the article, such as the number of front heads, scaling factors, etc., directly give fixed values, but do not specify the basis and rationality of the setting, nor do they evaluate the sensitivity of their value changes to model performance. Maybe try to add some details about hyperparameter settings.

2. The article mainly uses quantitative indicators to evaluate the effectiveness of the method. Perhaps several qualitative analysis examples can be tried to add to the appendix, such as showing the original output of the model under specific tasks and the output after using the method. This will more intuitively reflect the effectiveness of the proposed method.

**Ethical Concerns:**

["NO or VERY MINOR ethics concerns only"]

**Final Justification:**

I would like to thank the authors for the rebuttal, which has somehow tackled weakness. I will retain my original score.

Best regards,

**Limitations:**

yes

**Quality:**

3

**Strengths And Weaknesses:**

Strengths：

1. The paper has a rigorous structure and is highly readable.

2. The intervention is carried out at the reasoning stage, which does not require additional training, requires less code modification, has low deployment costs and is easier to integrate.

3. The author found that LVLM automatically switches between different paths according to the format of the question, and this observation led to the final "discriminative" and "generative" intervention strategies, with a complete logical closed loop, which not only analyzes the mechanism but also guides the strategy design.

Weaknesses：

1. The method proposed in the article is able to adaptively select the key attention head and apply the scaling factor, but the potential impact of attention head intervention on the original accurate output of the model in the absence of "illusion" scenarios is not evaluated. Perhaps it is possible to consider supplementing the model's output changes before and after intervention in the hallucination-free sample. To verify the safety and stability of this method.

2. In Sec.2.3, the label source of the set T_I2T is not specified in the text, which will cause readers to be unable to determine whether these labels are based on manual labels or automatically generated. It is recommended to add a detailed description of the annotation process of this data set in the article to improve the repeatability and credibility of the experiment.

---

> ### Author Rebuttal · Authors · 2025-07-31
>
> Thank you for your recognition of our paper and our framework AllPath. Below, we provide detailed responses to the weaknesses and questions you raised:
>
> > **Weakness 1:** The potential impact of attention head intervention on the original accurate output of the model in the absence of "illusion" scenarios is not evaluated.
>
> Thank you for your valuable feedback. To assess the robustness of our approach on originally correct cases, we analyze the POPE COCO benchmark. The table below shows the number of originally correct and incorrect cases and the changes after applying our method. Our results demonstrate that our method effectively suppresses hallucinations, while its impact on correct (non-hallucinated) cases is negligible, with a change of only 1.5%. This indicates that our method exhibits high stability and safety when applied to non-hallucinated inputs.
>
> | Before \ After | Hal.         | Non-Hal.     |
> | -------------- | ------------ | ------------ |
> | Non-Hal.       | 116  ( 1.5%) | 7378 (98.5%) |
>
> > **Weakness 2:** Label source of the set T_I2T In Sec. 2.3.
>
> The set $\mathcal{T}\_\text{I2T}$ consists of tokens corresponding to the first occurrence of each object mentioned during the question-answering process, either in the input text $P$ or in the generated response $R\_{\leq t}$ (Appendix B.1).  Specifically, for short-answer questions like POPE and MCQ-POPE, the selected tokens are from the input text, while for open-ended questions like CHAIR, they are drawn from the generated response. In our implementation, we use the same parser for the CHAIR evaluation to automatically extract objects and remove synonyms, without relying on external models or manual annotations.
>
> The rationale behind this design is that only the first mention of an object should attend directly to the image, while subsequent references can rely on prior textual context. We will include these details and clarifications in the final version of the paper.
>
> > **Question 1:** Details about hyperparameter settings.
>
> Thank you for your valuable feedback. We include ablation experiments for the head and scaling factor. The results, as shown in the table, indicate that some alternative hyperparameter settings even resulted in better performance on POPE, which shows that our method is effective and stable without extensive tuning.
>
> |        |         |            |            | ran. |      | pop. |      | adv. |      |
> | ------ | ------- | ---------- | ---------- | ---- | ---- | ---- | ---- | ---- | ---- |
> | $\xi$  | $\zeta$ | $\gamma^+$ | $\gamma^-$ | Acc. | F1   | Acc. | F1   | Acc. | F1   |
> | **20** | **10**  | **2.0**    | **0.0**    | 87.2 | 86.0 | 86.0 | 84.9 | 82.8 | 82.1 |
> | **10** | 10      | 2.0        | 0.0        | 87.3 | 86.2 | 86.0 | 85.0 | 82.6 | 82.0 |
> | **30** | 10      | 2.0        | 0.0        | 88.3 | 87.7 | 86.7 | 86.3 | 82.9 | 83.0 |
> | **40** | 10      | 2.0        | 0.0        | 88.1 | 87.9 | 85.9 | 86.0 | 81.2 | 82.2 |
> | 20     | **5**   | 2.0        | 0.0        | 86.2 | 84.8 | 85.8 | 84.6 | 82.7 | 81.9 |
> | 20     | **15**  | 2.0        | 0.0        | 88.2 | 87.4 | 86.6 | 85.9 | 83.0 | 82.8 |
> | 20     | **20**  | 2.0        | 0.0        | 88.1 | 87.3 | 86.7 | 86.0 | 83.0 | 82.8 |
> | 20     | 10      | **1.2**    | 0.0        | 85.6 | 85.3 | 85.3 | 84.1 | 82.3 | 81.5 |
> | 20     | 10      | **1.5**    | 0.0        | 86.9 | 85.7 | 85.6 | 84.5 | 82.5 | 81.7 |
> | 20     | 10      | **2.5**    | 0.0        | 86.8 | 85.6 | 86.3 | 85.4 | 83.1 | 82.5 |
> | 20     | 10      | 2.0        | **0.2**    | 87.1 | 85.9 | 85.8 | 84.7 | 82.5 | 81.8 |
> | 20     | 10      | 2.0        | **0.5**    | 87.0 | 85.8 | 85.6 | 84.5 | 82.2 | 81.5 |
> | 20     | 10      | 2.0        | **0.8**    | 86.8 | 85.5 | 85.4 | 84.2 | 82.0 | 81.3 |
>
> > **Question 2:** Qualitative analysis examples.
>
> Thank you for the suggestion. In response, we provide one specific qualitative example below. We will include qualitative examples in the appendix, showing the model's original and modified outputs for specific tasks, to better illustrate the effectiveness of our method.
>
> - **Image**: A dog is sitting on a bench. Behind the bench, beyond a fence, there are various flowers planted directly in the soil. There are **no potted plants** in the image.
> - **Prompt**: Please help me describe the image in detail.
> - **Baseline Response (hallucination at ‘potted plants’)**:
>
>     The image features a dog sitting on a bench, possibly a park bench, with its tongue out. The dog is wearing a red leash, and it appears to be enjoying its time outdoors. The bench is located near a fence, and there are several **potted plants** surrounding the area.
>
>     In addition to the main dog, there are two other dogs in the scene, one located near the left edge of the image and the other towards the right side. The presence of multiple dogs suggests that this might be a popular spot for pet owners to bring their pets for a walk or a rest.
>
> - **Our Method’s Response (no hallucination):**
>
>     The image features a dog sitting on a bench, which is located near a flower garden. The dog is wearing a red leash and appears to be enjoying its time outdoors. The dog is positioned in the middle of the bench, occupying a significant portion of the bench's length.
>
>     The flower garden is filled with various flowers, creating a beautiful and colorful garden. The flowers are scattered throughout the garden, with some flowers located closer to the dog and others further away. The garden creates a beautiful and serene environment for the dog to enjoy.
>
> - **Behavior of Our Identified Heads.** When baseline is generating ‘potted plants’:
>     - The heads with the **highest I2T (image-to-text) scores** focused on the flower region in the background, grounding the description visually.
>     - The heads with the **lowest T2T (text-to-text) scores** distributed attention relatively uniformly across general terms like “dog”, “fence”, and “bench.”
> - **Interpretation**:
>
>     This example demonstrates a case where the LVLM exhibited partial alignment on two pathways. Our method adaptively amplified both the image-to-text and text-to-text pathways, reinforcing useful information and effectively suppressing hallucination.

---

> > ### Comment · Reviewer_5FfF · 2025-08-03
> >
> > Thanks for addressing my concern. I would like to maintain my score.
> >
> > best,

---

> > > ### Author Response · Authors · 2025-08-03
> > >
> > > We appreciate your response and thank you for taking the time to review our work. We will incorporate your suggestions in the final version of the paper.

---

### Official Review · Reviewer_vdH4 · 2025-06-29

**Clarity:** 3
**Significance:** 2
**Originality:** 2
**Rating:** 4
**Confidence:** 4

**Summary:**

This paper focuses on mitigating large vision language model hallucinations in a head-intervented manner. The author first expands existing metrics (LPI scores) into a grouped verision (T2T, I2T), and proposes to use these metrics to analyze each head's role in model hallucinations. Based on score comparisons across different datasets with different formats, the author claims that this multi-path intervention consistently reduces hallucinations across different task formats.

**Questions:**

- For the open-ended QA, what model is used to judge whether an answer is a hallucination or not?
- I know this might be a wording issue, but why do the authors think that their proposed method is using a "causal path"? To me, I do not understand why the whole problem and proposed solution is relevant to causality.

**Ethical Concerns:**

["NO or VERY MINOR ethics concerns only"]

**Final Justification:**

After the rebuttal, my concerns about the theoretical intuition and comparisons with existing baselines have been addressed. Therefore, I raise my rating to weak acceptance to reflect this.

**Limitations:**

No. I think understanding LVLM hallucinations beyond existing benchmarks will be very interesting and important. I encourage the authors to explore more on this rather than focusing on a few existing datasets, which only cover a tiny aspect of Vision QA.

**Paper Formatting Concerns:**

NaN

**Quality:**

2

**Strengths And Weaknesses:**

### Strengths

- **Important Problem:** The paper addresses the timely and critical problem of hallucination in LVLMs, which is a significant barrier to their reliable deployment in real-world applications.
- **Practicality:** The proposed intervention is training-free, making it efficient and easy to apply to existing models without requiring expensive fine-tuning.
- **Writing and Orginization:** This paper is writing in a relatively clear way, making the understanding of its intuition and algorithm easy.

### Weaknesses

There are several major weaknesses that prevent this work from being accepted. Specifically,

- The central analysis lacks methodological rigor. While the authors claim to uncover the roles of different "causal-paths" and "heads", the proposed score metrics (T2T and I2T) are presented as simple heuristics. There is no comprehensive explanation—either theoretical or empirical—for why these specific definitions can reliably quantify a head's contribution to hallucination. For instance, modeling a head's contribution as the log-probability difference from its isolated addition to the residual stream fails to account for the complex, non-additive interactions between multiple heads. A more rigorous approach, such as one inspired by Shapley values, would be needed to properly attribute contributions in such a system. The authors need to demonstrate what verifiable properties their scores possess; for example, just as word embeddings exhibit semantic regularities (e.g., king - man + woman ≈ queen), the authors should show what predictable, meaningful properties their scores hold.
- The explanatory sections (3.1 and 3.2) are not self-justified and fail to build a convincing narrative. The paper frequently uses vague and anthropomorphic language to describe its findings, such as heads "contributing," "differing," or models being "smarter" (L69-78), instead of presenting precise, falsifiable claims. The term "identified heads" is used throughout, but there is no clear principle for *why* certain heads are selected over others beyond their ranking on an unvalidated metric. As a result, the analysis does not convincingly support the intervention that follows.
- If I understand correctly, the whole algorithm for reducing hallucinations is basically a simple reweight of heads (L216). Were the proposed method outperform other training-free methods, I can ignore the novelty issue. Unfortunately, there are lots of training-free methods such as [1], that the authors do not compare with. Table 4 in [1] shows that for LLaVA1.5, their F1 score and Accuracy outperform the propose method on 2/3 setttings on POPE dataset. This paper is released in 2024 and its follow-ups have better performance. Given the limited novelty, I do not think that the paper meets the acceptance threshold unless their performance can be significantly better than existing approaches.

[1] Reducing hallucinations in large vision-language models via latent space steering

---

> ### Author Rebuttal · Authors · 2025-07-31
>
> We sincerely appreciate your time and effort. Below, we respond to the weaknesses and questions raised, offering clarifications and justifications to further highlight the novelty and effectiveness of our framework:
>
> > **Weakness 1 & 2:** There is no comprehensive explanation for why these specific definitions can reliably quantify a head's contribution to hallucination, and there is no clear principle for why certain heads are selected over others beyond their ranking on an unvalidated metric.
>
> ### **Theoretical rationale behind our framework**
>
> Our rationale behind the overall framework and the design of the two metrics is as follows:
>
> 1. **Rationale of the Overall Framework.** Due to the causal attention mechanism of LVLMs, generation is influenced by both the image-to-text and text-to-text pathways. Therefore, we aim to analyze how each pathway contributes to hallucination and mitigate hallucinations by intervening in both paths. Within these two pathways, there are two distinct types of atomic paths, one is the image-to-text path, and the other is the text-to-text path. So we propose two separate metrics to identify the hallucinatory heads associated within each pathways.
> 2. **Rationale Behind the Hallucination Metrics**.
>     - **Text-to-Text Score**. It has been suggested that unembedding the intermediate hidden states of LVLMs can approximate the model’s internal beliefs or predictions at each step [1]. Based on this, our Text-to-Text (T2T) metric uses these approximations to assess the influence of individual attention heads on generating non-hallucinated versus hallucinated responses. To achieve this, we employ the log-probability increase score to estimate the contribution of each individual head to hallucinatory and non-hallucinatory outputs. This enables us to identify attention heads that suppress or contribute to hallucinations.
>     - **Image-to-Text Score**. It has been proved that enhancing image-related attention helps mitigate hallucinations [2]. Inspired by this, we propose the I2T metric to quantify each attention head's contribution to the model's focus on image features, allowing us to identify attention heads linked to hallucinations.
>
> **Furthermore, based on the transferability and strong generalization of the heads identified by our metrics, this in turn demonstrates the effectiveness of our heads.**
>
> - The heads identified on the POPE COCO adversarial subset and the MCQ-POPE COCO adversarial subset can be applied respectively across all POPE and MCQ-POPE tasks (see Table 1 in the paper);
> - Attention heads identified on POPE can be directly leveraged to improve performance on MME (see Table 2 in the paper).
>
> Such transferability is neither trivial nor heuristic.
>
> **Regarding other metrics and joint interactions.** Notably, we have considered common techniques for interpretation like gradient-based and zero-out methods; however, these approaches are computationally expensive and may not scale efficiently in practice. On the other hand, the joint interaction between attention heads and model responses is a complex issue within the domain of interpretability, and it falls beyond the scope of our current work. Our focus is on identifying the key attention heads that influence hallucinations, aiming to effectively mitigate their occurrence.
>
> ### **Empirical explanation of our framework**
>
> Based on your suggestions, we present empirical evidence demonstrating the effectiveness of our metrics:
>
> - **Text-to-Text Score**. We demonstrate the effectiveness of T2T Heads on the hallucination benchmark in Table 3. Additionally, we conduct an experiment where we amplify the identified heads. The results, shown in the table below, reveal that amplifying the top 20 heads (with the highest T2T scores) significantly reduces hallucinations, improving accuracy by at least 3.2% across each subset. Conversely, amplifying the bottom 20 heads leads to a sharp decline in accuracy, dropping by at least 44.7%. This indicates a strong correlation between T2T heads and hallucinations.
>
>     | LLaVA 1.5 7B           | ran. |      | pop. |      | adv. |      |
>     | ---------------------- | ---- | ---- | ---- | ---- | ---- | ---- |
>     |                        | Acc. | F1   | Acc. | F1   | Acc. | F1   |
>     | Vanilla                | 85.1 | 83.7 | 83.7 | 82.4 | 80.9 | 80.0 |
>     | T2T top-20 heads 8×    | 88.6 | 87.7 | 88.0 | 87.2 | 84.1 | 83.4 |
>     | T2T bottom-20 heads 8× | 40.4 | 39.8 | 35.8 | 37.0 | 33.7 | 36.7 |
>
> - **Image-to-Text Score**. We provide visual evidence for this metric in Fig. 4. The extracted I2T heads focus on object-relevant regions of the image, while using all heads results in more dispersed attention, indicating that our selected heads attend to query- and content-related visual information. We will include attention maps for all heads in the final version for comparison. The right side of Fig 4 further shows that I2T heads focus more on the image than averaging all heads, with these heads being highly correlated with hallucinations. Finally, the performance boost in Table 3 validates that the extracted heads effectively target hallucinations, with accuracy improving by at least 1.3%.
>
> > **Weakness 3:** Concerns about the novelty and performance.
>
> **The novelty of our method is multi-fold as follows:**
> 1. While our method involves reweighting, **its core contribution is the identification of key heads in different types of paths**, which enables targeted reweighting to achieve substantial performance improvements. This is something that simple reweighting methods do not achieve.
> 2. To the best of our knowledge, we are the first to investigate the impact of different pathways on hallucination, and we discover that **the importance of these pathways varies significantly across task formats**, offering several novel insights.
> 3. Furthermore, we are the first to introduce **pathway-specific hallucination metrics for I2T and T2T heads**, and we **empirically demonstrate their effectiveness**.
>
> **On comparison with [1] and performance.** Please note that [1] uses a different prompt format and decoding setup: they adopt prompt “Is there a [object] in the image?” and use beam search (beam size = 5). In contrast, we follow the official LLaVA evaluation setup, where prompts are appended with “Please answer this question with one word.” and does not uses beam search when evaluating POPE following VCD [2] and ICD [3].
>
> We have reproduced the performance of the VTI [1] under our setup using their official code. The results are shown in the table below. As seen, VTI underperforms both the vanilla baseline and our proposed method across all splits of the POPE benchmark. What’s more, on the CHAIR benchmark, as reported in respective papers, our method outperforms VTI by 9.2 points on $CHAIR_S$ and by 3.9 points on $CHAIR_I$, under similar baseline performances.
>
> | LLaVA 1.5 7B | ran. |      | pop. |      | adv. |      |
> | ------------ | ---- | ---- | ---- | ---- | ---- | ---- |
> |              | Acc. | F1   | Acc. | F1   | Acc. | F1   |
> | Vanilla      | 85.1 | 83.7 | 83.7 | 82.4 | 80.9 | 80.0 |
> | VTI          | 83.2 | 80.9 | 81.8 | 79.5 | 79.0 | 77.1 |
> | Ours         | 87.2 | 86.0 | 86.0 | 84.9 | 82.8 | 82.1 |
>
> | LLaVA 1.5 7B | $CHAIR_S$ | $CHAIR_I$ |
> | ------------ | --------- | --------- |
> | VTI          | 35.8      | 11.1      |
> | Ours         | 26.6      | 7.2       |
>
> [2] Mitigating Object Hallucinations in Large Vision-Language Models through Visual Contrastive Decoding
>
> [3] Mitigating Hallucinations in Large Vision-Language Models with Instruction Contrastive Decoding
>
> > **Question 1:** For the open-ended QA, what model is used to judge whether an answer is a hallucination or not?
>
> Our experiments are conducted on the CHAIR benchmark, using its official evaluation protocol for hallucination, which is a widely adopted standard in the field. Specifically, the CHAIR parser extracts object mentions from the generated captions and maps them to the 80 object categories defined in the COCO dataset. These extracted objects are then compared with the GT annotations to determine whether an object is hallucinated. This process does not involve any language models.
>
> > **Question 2:** Why the whole problem and proposed solution are relevant to causality?
>
> We use the term **“**causal path**”** to refer to components in the model that have a causal influence on the final output, specifically on hallucination. This is not meant in the strict Pearlian sense, but rather follows an intervention-based perspective: when we intervene on a path (e.g., remove it), the output changes accordingly.
>
> Importantly, while all pathways exist in every generation, their causal effect on hallucination varies by format. For example, in POPE, the image-to-output-text path has minimal causal influence on the final answer, while in CHAIR, this path has a significant causal effect. This difference supports our claim that causal paths are format-dependent, as illustrated in Fig. 5 (right) — the former rely on the Image-Prompt-Response pathway and the latter introduces an additional path: Input → Response.

---

> > ### Comment · Reviewer_vdH4 · 2025-08-06
> >
> > The reviewer thanks the authors for their response. Most of my concerns have been addressed, and I will raise my rating accordingly.

---

> > > ### Author Response · Authors · 2025-08-07
> > >
> > > Thank you for your feedback! We are pleased to have addressed most of your concerns. Should you have any remaining questions that have not been fully resolved, if so, please do not hesitate to contact us. We will make every effort to respond promptly to provide further clarification or address any additional questions you may have. We sincerely appreciate you once again for your valuable suggestions, recognition, and your decision to raise the score!

---

> ### Author Response · Authors · 2025-08-06
>
> Dear Reviewer vdH4,
>
> As the discussion period is ending in less than 3 days, we would greatly appreciate any further comments or suggestions on our work from you to see if our responses solve your concerns.
> Thank you very much for your time and consideration.
>
> Sincerely,
> The Authors

---

### Official Review · Reviewer_JSpS · 2025-07-01

**Clarity:** 2
**Significance:** 2
**Originality:** 3
**Rating:** 5
**Confidence:** 3

**Summary:**

This paper introduces AllPath, an intervention framework designed to mitigate hallucinations in Vision-Language Models (LVLM) across diverse alignment formats. The framework employs a multi-path approach that analyzes three critical causal pathways: image2input-text, image2output-text, and text2text interactions within LVLMs.

The key innovation is a novel method for identifying critical attention heads along these pathways that contribute to hallucination generation. Empirical findings reveal that LVLMs dynamically adapt their pathway usage depending on specific question-answer formats.

Building on this findings, the authors propose a targeted intervention method that selects optimal pathways and attention heads based on question type. The framework's effectiveness is rigorously validated through extensive experiments across multiple benchmarks, including POPE, MCQ-POPE, CHAIR, and MME, demonstrating consistent performance improvements over baseline methods in various alignment scenarios.

**Questions:**

- How sensitive is the AllPath method to the choice of hyper-parameters, particularly the number of selected heads?

- Have you tested your approach on larger VLMs? Is the head issue present in larger VLMs?

- The paper mentions that different question formats rely on distinct causal
pathways. Can you expand on it? It is unclear what is the exact difference here.

**Ethical Concerns:**

["NO or VERY MINOR ethics concerns only"]

**Final Justification:**

The rebuttal and the additional ablations are informative. The additional experiments clearly provide evidence that this method can be applied on larger and more powerful VLMs. I'm raising my score.

**Limitations:**

The authors have adequately addressed the limitations of their work. They acknowledge that their primary analysis focuses on one LVLM and provide additional results for other models in the appendix.

**Paper Formatting Concerns:**

No major concerns

**Quality:**

3

**Strengths And Weaknesses:**

**Strengths**

This paper introduces a comprehensive framework for reducing hallucinations in large vision-language models by analyzing multiple causal pathways, including how images and text interact within the model.

The authors provide novel insights into how these models use different pathways depending on the type of question asked. The proposed method achieves strong and consistent improvements across several standard benchmarks and question types, demonstrating its effectiveness in practice.

**Weaknesses**

Despite the experimental results, the paper does not provide a theoretical explanation for why the framework works.
Additionally, most experiments are conducted on a single model (LLaVA-v1.5-7B), so it is unclear how well the method would perform on other models, especially larger ones. Broader testing on different models would help confirm the general usefulness of the approach.

---

> ### Author Rebuttal · Authors · 2025-07-31
>
> Thank you for recognizing our causal pathway analysis and framework. Below, we provide detailed replies to each of the weaknesses and questions:
>
> > **Weakness 1:** The paper does not provide a theoretical explanation for why the framework works.
>
> Thanks for your feedback and suggestion. Our rationale behind the overall framework, the design of the two metrics, and the intervention on attention heads to mitigate hallucinations is as follows:
>
> 1. **Rationale of the Overall Framework.** Due to the causal attention mechanism of LVLMs, generation is influenced by both the image-to-text and text-to-text pathways. Therefore, we aim to analyze how each pathway contributes to hallucination and mitigate hallucinations by intervening in both paths. Within these two pathways, there are two distinct types of atomic paths, one is the image-to-text path, and the other is the text-to-text path. So we propose two separate metrics to identify the hallucinatory heads associated within each pathways.
> 2. **Rationale Behind the Hallucination Metrics**:
>     - **Text-to-Text Score**. It has been suggested that unembedding the intermediate hidden states of LVLMs can approximate the model’s internal beliefs or predictions at each step [1]. Based on this, our Text-to-Text (T2T) metric uses these approximations to assess the influence of individual attention heads on generating non-hallucinated versus hallucinated responses. To achieve this, we employ the log-probability increase score to estimate the contribution of each individual head to hallucinatory and non-hallucinatory outputs. This enables us to identify attention heads that suppress or contribute to hallucinations.
>     - **Image-to-Text Score**. It has been proved that enhancing image-related attention helps mitigate hallucinations [2]. Inspired by this, we propose the I2T metric to quantify each attention head's contribution to the model's focus on image features, allowing us to identify attention heads linked to hallucinations.
> 3. **Rationale behind our AllPath for hallucination mitigation** is that it precisely suppress attention heads that contribute to hallucinations while enhancing those that help the model produce correct outputs. Since we have already identified hallucination-related heads, hallucinations can be effectively mitigated through a simple strategy of reweighting these heads. Our quantitative results further validate that our identified heads play a meaningful and consistent role in hallucinations.
>
> [1] Understanding Multimodal LLMs: The Mechanistic Interpretability of LLaVA in Visual Question Answering
> [2] Paying More Attention to Image: A Training-Free Method for Alleviating Hallucination in LVLMs
>
> > **Weakness 2 & Question 2:** Test on other models, especially larger ones.
>
> Thank you for your valuable comment. We include results on Qwen VL, the newer and stronger model from the same series, Qwen2.5-VL-7B, and the much larger Qwen2.5-VL-72B, on POPE to further evaluate the generalizability of our method across models of different scales. Since Qwen2.5-VL-72B has five times as many heads as LLaVA, we accordingly increased the number of heads we used by a factor of five.
>
> | Qwen-VL  | ran. |      | pop. |      | adv. |      |
> | -------- | ---- | ---- | ---- | ---- | ---- | ---- |
> |          | Acc. | F1   | Acc. | F1   | Acc. | F1   |
> | Vanilla  | 84.5 | 82.3 | 83.4 | 81.4 | 80.5 | 78.9 |
> | Ours     | 86.6 | 85.3 | 85.6 | 84.3 | 82.4 | 81.5 |
> | $\Delta$ | +2.1 | +3.0 | +2.2 | +2.9 | +1.9 | +2.6 |
>
> | Qwen2.5-VL-7B | ran. |      | pop. |      | adv. |      |
> | ------------- | ---- | ---- | ---- | ---- | ---- | ---- |
> |               | Acc. | F1   | Acc. | F1   | Acc. | F1   |
> | Vanilla       | 87.6 | 86.0 | 86.6 | 85.1 | 85.5 | 84.0 |
> | Ours          | 90.6 | 90.0 | 88.6 | 88.1 | 86.6 | 86.3 |
> | $\Delta$      | +3.0 | +4.0 | +2.0 | +3.0 | +1.1 | +2.3 |
>
> | Qwen2.5-VL-72B | ran. |      | pop. |      | adv. |      |
> | -------------- | ---- | ---- | ---- | ---- | ---- | ---- |
> |                | Acc. | F1   | Acc. | F1   | Acc. | F1   |
> | Vanilla        | 87.7 | 86.2 | 86.6 | 85.1 | 85.6 | 84.1 |
> | Ours           | 90.7 | 90.2 | 88.5 | 88.1 | 85.7 | 85.7 |
> | $\Delta$       | +3.0 | +4.0 | +1.9 | +3.0 | +0.1 | +1.6 |
>
> > **Question 1:** How sensitive is the AllPath method to the choice of hyper-parameters.
>
> Thank you for your question. We conduct additional ablation studies on the hyper-parameters on POPE based on LLaVA 1.5 7B. The results are shown in the table below. It illustrates that the performance of our method is not sensitive to the choice of this hyper-parameter. Notably, alternative hyper-parameter settings resulted in better performance on POPE, demonstrating the robustness of our approach. Rather than tuning for a specific dataset, we selected a single set of hyper-parameters that works well across various discriminative tasks. This shows that our method is effective and stable without extensive tuning. We will include these results in the revised paper.
>
> |        |         |            |            | ran. |      | pop. |      | adv. |      |
> | ------ | ------- | ---------- | ---------- | ---- | ---- | ---- | ---- | ---- | ---- |
> | $\xi$  | $\zeta$ | $\gamma^+$ | $\gamma^-$ | Acc. | F1   | Acc. | F1   | Acc. | F1   |
> | **20** | **10**  | **2.0**    | **0.0**    | 87.2 | 86.0 | 86.0 | 84.9 | 82.8 | 82.1 |
> | **10** | 10      | 2.0        | 0.0        | 87.3 | 86.2 | 86.0 | 85.0 | 82.6 | 82.0 |
> | **30** | 10      | 2.0        | 0.0        | 88.3 | 87.7 | 86.7 | 86.3 | 82.9 | 83.0 |
> | **40** | 10      | 2.0        | 0.0        | 88.1 | 87.9 | 85.9 | 86.0 | 81.2 | 82.2 |
> | 20     | **5**   | 2.0        | 0.0        | 86.2 | 84.8 | 85.8 | 84.6 | 82.7 | 81.9 |
> | 20     | **15**  | 2.0        | 0.0        | 88.2 | 87.4 | 86.6 | 85.9 | 83.0 | 82.8 |
> | 20     | **20**  | 2.0        | 0.0        | 88.1 | 87.3 | 86.7 | 86.0 | 83.0 | 82.8 |
> | 20     | 10      | **1.2**    | 0.0        | 85.6 | 85.3 | 85.3 | 84.1 | 82.3 | 81.5 |
> | 20     | 10      | **1.5**    | 0.0        | 86.9 | 85.7 | 85.6 | 84.5 | 82.5 | 81.7 |
> | 20     | 10      | **2.5**    | 0.0        | 86.8 | 85.6 | 86.3 | 85.4 | 83.1 | 82.5 |
> | 20     | 10      | 2.0        | **0.2**    | 87.1 | 85.9 | 85.8 | 84.7 | 82.5 | 81.8 |
> | 20     | 10      | 2.0        | **0.5**    | 87.0 | 85.8 | 85.6 | 84.5 | 82.2 | 81.5 |
> | 20     | 10      | 2.0        | **0.8**    | 86.8 | 85.5 | 85.4 | 84.2 | 82.0 | 81.3 |
>
> > **Question 3:** Differences in how different question formats rely on distinct causal pathways.
>
> Causal paths including image-to-input-text, image-to-output-text, and input-text-to-output-text are always present across LVLMs (via casual attention mechanisms), but their significance varies with the question format.
>
> For discriminative tasks like POPE, masking the image-to-output-text path results in a minor performance drop (Fig. 5 left in paper), which means that POPE requires little or much less the image-to-output-text path. However, in open-ended tasks such as CHAIR, masking the same path causes a more significant decline, indicating that this path plays a more critical role in CHAIR. Taken together, these findings indicate distinct causal paths for each task: POPE follows Input → Prompt → Response, while CHAIR introduces an additional path: Input → Response (Fig. 5, right). These have been discussed in Section 3.2 and Fig. 5 in our paper.

---

> > ### Comment · Reviewer_JSpS · 2025-08-04
> >
> > Thank you for the rebuttal and the additional experiments. I'm raising my score.

---

> > > ### Author Response · Authors · 2025-08-04
> > >
> > > Thank you for the encouraging feedback and valuable suggestions! We will revise the paper accordingly by adding the experiments on Qwen-VL, Qwen2.5-VL-7B, and Qwen2.5-VL-72B, and by conducting ablation studies on the hyperparameters. We truly appreciate your input!

---

### Official Review · Reviewer_Zj3C · 2025-07-09

**Clarity:** 3
**Significance:** 3
**Originality:** 3
**Rating:** 5
**Confidence:** 3

**Summary:**

The paper argues that hallucinations in Large Vision-Language Models (LVLMs) don't stem from a single error source but rather emerge from complex interactions between multiple information pathways within the model's architecture. The authors develop a comprehensive intervention framework that simultaneously targets both image-to-output-text and image-to-input-text-to-output-text causal pathways. They establish importance scores for individual attention heads across text-to-text and image-to-text pathways, revealing that heads in different pathways behave distinctly and vary depending on question formats. Through targeted intervention of attention heads—both suppressing those associated with hallucination promotion and enhancing those linked to hallucination suppression across image-to-input-text and image-to-input-to-output-text pathways—they achieve approximately 2% reductions in hallucination rates compared to baseline methods across multiple benchmarks. Ablation studies demonstrate that both causal pathways are essential for optimal performance and a single universal head configuration proves effective across different datasets.

**Questions:**

- Could you please elaborate on how the analysis on attention heads adds beyond the attention analysis in [1]?
- Could you illustrate qualitative examples on where the intervention succeeds and fails?
- There is a typo in line 162: `Image-to-input-text causal path is different from image-to-input-text causal path` – one of the `image-to-input-text` has to be corrected to `image-to-output-text`

**Ethical Concerns:**

["NO or VERY MINOR ethics concerns only"]

**Final Justification:**

All issues raised have been resolved. I am convinced that this submission makes unique contributions that differentiate it from the prior work mentioned in [1], and the qualitative examples effectively demonstrate when and how the proposed mechanism performs well.
Regarding the failure scenario identified in the author rebuttal, I have suggested that the authors discuss it in the Limitations section to provide readers with a complete understanding of the method's boundaries.

I will maintain my score as Accept.

**Limitations:**

Yes

**Paper Formatting Concerns:**

The only issue I noticed is that section headings are written in Title Case instead of the required lower case specified in the formatting guidelines.

**Quality:**

4

**Strengths And Weaknesses:**

**Strengths**
- Jointly addressing causal pathways to treat hallucination outperforms prior single-path fixes, which is intuitive and engineered cleverly.
- The analysis is granular and informs the method design well.
- The method and evaluation setup is comprehensive. The authors probe both direct and indirect visual pathways across diverse question formats, two LVLM architectures (LLaVA v1.5 7B and Qwen-VL-Chat), and strong baselines, with methodology clearly documented.
- The gains are consistent and high on multiple benchmarks.

**Weaknesses**
- The paper’s finding that the image-to-input-text and image-to-output-text pathways are governed by different sets of heads are similarly explored in previous work. [1] also separately analyzes the information flow from image-to-query tokens and from image-to-generated tokens. The idea of analyzing distinct stages of visual information processing was a known direction in the field, making the analysis aspect of the paper’s contribution incremental.

[1] Kaduri et al., What's in the Image? A Deep-Dive into the Vision of Vision Language Models. *CVPR 2025*.

---

> ### Author Rebuttal · Authors · 2025-07-31
>
> Thank you for recognizing our findings on the causal pathways as well as our method and evaluation setup. We especially appreciate that you insightfully pointed out prior work [1]. Below, we provide detailed responses to the weaknesses and questions you raised:
>
> > **Weakness 1 & Question 1:** Analysis on attention heads adds beyond the attention analysis in [1].
>
> Thank you for your insightful comments and questions. While our findings share some similarities with those in [1], our approach advances beyond [1] in several important aspects:
>
> 1. **Broader Scope Enables the First Exploration of Hallucination in Alignment Formats.** Unlike [1], which focuses solely on the image captioning task, our study evaluate hallucination in different question formats, including yes/no tasks, multiple-choice tasks, and captioning tasks. This enables us to identify our core finding: **the importance of different pathways varies significantly across task formats. This is a phenomenon not previously observed in previous studies.** This explains why existing methods struggle to achieve consistent improvements across all benchmarks.
> 2. **A Finer-Grained Metric (Our Identification of Attention Heads in Each Layer vs. Averaged Attention Weight Across Layers in [1]) Enables In-Depth Analysis and Yields Several Novel Insights.** While [1]’s primarily analysis rely on the average attention weight across layers, we focus on identifying key attention heads in each layer, which enables a more fine-grained and interpretable understanding of the underlying mechanisms. This methodological refinement leads to **several novel insights that go beyond [1]:**
>     1. **Although both yes/no and multiple-choice tasks rely on the Image-Prompt-Response pathway, they rely on very different attention heads** (see Fig. 3 the third column in paper). This reveals that even when different format of questions share a same pathway, the actual cross-modal attention heads they rely on are not the same, which advances beyond [1]’s general observations about the I-P pathway. What’s more, our method not only identifies these heads precisely, but also validates their significance via performance gains in hallucination mitigation experiments.
>     2. By identifying heads, we further uncover the detailed mechanisms in I2T and T2T interactions. While [1] observes that both input and output texts pay attention to the image (as we do in Fig. 4), **we go further by showing that the output text specifically attends to object-level keywords in the input text**, which is more obvious in yes/no and multiple-choice tasks. **These patterns are more obvious at the level of heads than in average attention.** Due to the rebuttal format's limitations, we describe this phenomenon in text, and we will provide the corresponding visualizations in the revised version of our paper.
>     3. Additionally, our head-level analysis yields other valuable insights. For example, we reaffirm that the importance of paying attention to image in mitigating hallucinations. These further supporting the contribution of our work.
> 3. **Head-Level Analysis Enables Practical Utility in Hallucination Mitigation.** Because we can accurately identify hallucinatory attention heads, we are able to use these heads to effectively mitigate hallucinations and enhance performance. In contrast, [1] offers some high-level interpretability of the model’s behavior, but its findings offer less direct utility for improving the LVLM itself.
>
> > **Question 2:** Qualitative examples on where the intervention succeeds and fails.
>
> Thank you for your question. Our method is particularly effective when the LVLM demonstrates correct reasoning even along just one of the two pathways (e.g., image-to-text or text-to-text). In such scenarios, our intervention adaptively reinforces the pathway to make it more reliable, thereby reducing hallucination. However, our method is less effective when both pathways in the LVLM exhibit substantial errors, making it difficult for the intervention—despite targeting certain heads—to fully correct the output. Below, we present one successful case and one failure case to illustrate this:
>
> - **Success Case**
>     - **Image**: A dog is sitting on a bench. Behind the bench, beyond a fence, there are various flowers planted directly in the soil. There are **no potted plants** in the image.
>     - **Prompt**: Please help me describe the image in detail.
>     - **Baseline Response (hallucination at ‘potted plants’)**:
>
>         The image features a dog sitting on a bench, possibly a park bench, with its tongue out. The dog is wearing a red leash, and it appears to be enjoying its time outdoors. The bench is located near a fence, and there are several **potted plants** surrounding the area.
>
>         In addition to the main dog, there are two other dogs in the scene, one located near the left edge of the image and the other towards the right side. The presence of multiple dogs suggests that this might be a popular spot for pet owners to bring their pets for a walk or a rest.
>
>     - **Our Method’s Response (no hallucination):**
>
>         The image features a dog sitting on a bench, which is located near a flower garden. The dog is wearing a red leash and appears to be enjoying its time outdoors. The dog is positioned in the middle of the bench, occupying a significant portion of the bench's length.
>
>         The flower garden is filled with various flowers, creating a beautiful and colorful garden. The flowers are scattered throughout the garden, with some flowers located closer to the dog and others further away. The garden creates a beautiful and serene environment for the dog to enjoy.
>
>     - **Behavior of Our Identified Heads.** When baseline is generating ‘potted plants’:
>         - The heads with the **highest I2T (image-to-text) scores** focused on the flower region in the background, grounding the description visually.
>         - The heads with the **lowest T2T (text-to-text) scores** distributed attention relatively uniformly across general terms like “dog”, “fence”, and “bench.”
>     - **Interpretation**:
>
>         This example demonstrates a case where the LVLM exhibited partial alignment on two pathways. Our method adaptively amplified both the image-to-text and text-to-text pathways, reinforcing useful information and effectively suppressing hallucination.
>
> - **Failure case**:
>     - **Image**: A kitchen counter with a blender and miscellaneous clutter. There are no bowls or cups present.
>     - **Prompt**: Please help me describe the image in detail.
>     - **Baseline Response (hallucination at ‘potted plants’)**:
>
>         The image features a kitchen counter with a blender sitting on it. The blender is filled with a mixture, possibly a smoothie or a milkshake, and is ready to be blended. The counter is also covered in a brown substance, which could be a result of the blending process or a spill.
>
>         In addition to the blender, there are two **bowls** on the counter. One bowl is located on the left side of the blender, while the other is positioned on the right side. The kitchen counter appears to be a bit messy, with the blender and the brown substance taking up a significant portion of the space.
>
>     - **Our Method’s Response (hallucination at ‘cup’):**
>
>         The image features a kitchen counter with a blender sitting on it. The blender is filled with a mixture, possibly a smoothie or a smoothie-like substance. The blender is turned on, and the blender **cup** is filled with the mixture. The counter is also covered in a bit of mess, with a few drops of liquid spread across the counter.
>
>     - **Behavior of Our Identified Heads.** When baseline is generating ‘bowls’:
>         - The heads with the **highest I2T (image-to-text) scores** failed to sufficiently attend to the cluttered counter area in the image where no bowls could be verified.
>         - The heads with the **lowest T2T (text-to-text) scores** still maintained strong attention to token ‘blender’, suggesting that text priors dominated the response generation.
>     - **Interpretation**:
>
>         In this case, both pathways exhibited substantial issues: the image-to-text heads failed to focus on the visually relevant regions, while the text-to-text heads remained overly influenced by strong textual priors ("blender"). Since neither pathway is reliable, our intervention could not mitigate this hallucination.
>
> We will include these analysis and cases in the final version of our paper.
>
> > **Question 3 & Paper formatting concerns.**
>
> Thank you for pointing out the typos and formatting issues. We will correct them carefully in the final version.

---

> > ### Comment · Reviewer_Zj3C · 2025-08-03
> >
> > Thank you for the comprehensive and detailed response. I am convinced that this submission makes unique contributions that differentiate it from the prior work mentioned in [1], and the qualitative examples effectively demonstrate when and how the proposed mechanism performs well.
> >
> > Regarding the failure case, it would be helpful to include in the Limitations section that the proposed intervention method fails when both the I2T and T2T pathways are unreliable, e.g., due to strong textual priors.

---

> > > ### Author Response · Authors · 2025-08-03
> > >
> > > We sincerely thank the reviewer for the encouraging feedback and valuable suggestion! We will revise the paper accordingly by including a more detailed discussion of [1], adding qualitative results, and incorporating the suggested failure case into the Limitations section to provide a more comprehensive discussion.

---

### Note · Authors · 2025-08-15

Dear ACs and Reviewers,

We sincerely thank the ACs for their guidance and all reviewers for their constructive feedback. We are encouraged by the reviewers’ positive recognition of our main contributions:

- **Comprehensive Analysis with Novel Insights into Multi-Path Mechanisms.** We are the first to identify multi-path mechanisms underlying hallucinations in LVLMs and reveal that different question formats engage distinct causal pathways. Reviewers commend our granular analysis (Zj3C), comprehensive framework and novel insights (JSpS), logically complete analytical loop (5FfF), and is well-aligned with method design (Zj3C, 5FfF).
- **Intuitive, Practically Applicable Method.** Building on our multi-path mechanism, we propose a training-free intervention that can be seamlessly applied to diverse LVLMs. Reviewers highlight its intuitive design (Zj3C), practicality and training-free nature (vdH4, 5FfF), and ease of integration without extra cost (5FfF).
- **Extensive Experiments Demonstrating Consistent Improvement.** We evaluate our method across multiple models (LLaVA-v1.5-7B, Qwen-VL-Chat, Qwen2.5-VL-7B and 72B) and question formats (Yes/No, Multiple-Choice, Open-ended), showing consistent gains. Reviewers appreciate the comprehensiveness (Zj3C) and consistently strong performance over prior single-path solutions (Zj3C, JSpS).
- **Clear and Readable Presentation.** Our writing is noted as clear, logically structured, and easy to follow (vdH4, 5FfF).

In response to the reviewers’ insightful comments, we have made the following revisions:

- Providing **additional qualitative results and analyses** to highlight both the strengths and failure modes of our framework (Zj3C, 5FfF).
- **Clarifying key distinctions** between our approach and prior work (Zj3C).
- Offering **theoretical justification and restate empirical evidence** supporting our framework (JSpS, vdH4).
- Validating our method on Qwen-2.5-7B and 72B, demonstrating the **generalizability** of our approach across more and larger LVLMs (JSpS).
- Conducting extensive studies to confirm that our method is **robust to hyperparameters, and has negligible negative impacts** (JSpS, 5FfF).

We thank the reviewers for their constructive feedback and positive assessments. We appreciate their acknowledgement that our responses satisfactorily addressed their concerns, and their insights have further enhanced the rigor and relevance of the study.

Best regards,

The Authors

---

### Decision · Program_Chairs · 2025-09-17

**Decision:**

Accept (poster)

**Comment:**

This paper presents a training-free, head-level intervention framework that mitigates LVLM hallucinations by jointly targeting image-to-text and text-to-text causal pathways and shows these pathways shift with question format. Reviewers praised the novelty, consistent 2-4% gains across POPE, CHAIR and MME on LLaVA-v1.5, Qwen-VL and 72B Qwen2.5-VL, and the practical plug-and-play design; concerns about theoretical grounding, single-model focus, hyper-parameter sensitivity and comparison with prior steering methods were largely resolved through additional ablations, larger-model results and qualitative examples. Based on the above reason, this paper is recommended to be accpted.